# Electrochemical Deposition of Silicon: A Critical Review of Electrolyte Systems for Industrial Implementation

**DOI:** 10.3390/ma18174009

**Published:** 2025-08-27

**Authors:** Gevorg Abramkin, Srecko Stopic, Andrey Yasinskiy, Alexander Birich, Bernd Friedrich

**Affiliations:** IME Process Metallurgy and Metal Recycling, RWTH Aachen University, 52056 Aachen, Germany; aiasinskii@metallurgie.rwth-aachen.de (A.Y.); abirich@ime-aachen.de (A.B.); bfriedrich@metallurgie.rwth-aachen.de (B.F.)

**Keywords:** silicon electrodeposition, molten salts, electrolyte composition, sustainable silicon production

## Abstract

Electrochemical deposition of silicon is considered a promising alternative to conventional high-temperature and high-emission methods of silicon production. This review analyzes the current state of research on electrolyte systems used for silicon electrodeposition, with a particular focus on their potential for industrial-scale application. These systems are evaluated based on key characteristics relevant to such implementation, including silicon precursor solubility, electrical conductivity, applicable current density, and behavior under process conditions. The study evaluates fluoride-based, chloride-based, mixed halide, and organic electrolyte systems based on key criteria, including conductivity, chemical stability, silicon precursor solubility, temperature range, and ease of product purification. Fluoride-based melts offer high current densities (up to 2 A/cm^2^) and effective SiO_2_ dissolution but operate at high temperatures (550–1300 °C) and suffer from hygroscopicity. Chloride systems exhibit lower operating temperatures (300–1000 °C) and better water solubility but lack compatibility with common silicon sources. Mixed fluoride–chloride electrolytes emerge as the most promising option, combining high performance with improved practicality; they operate at 600–850 °C and current densities up to ~1.5 A/cm^2^. Additional focus is placed on the impact of substrate materials and on unresolved questions related to reaction reversibility, kinetic mechanisms, and the influence of electrolyte composition. The review concludes that further fundamental studies are needed to optimize electrolyte design and enable the transition from laboratory-scale research to industrial implementation.

## 1. Introduction

Electrodeposition of silicon has emerged as a promising alternative to conventional silicon production methods, offering significant potential for reducing carbon dioxide emissions. Compared to the conventional metallurgical and Siemens processes, the electrochemical route requires substantially less energy and generates considerably lower direct CO_2_ emissions (Table 1) [1]. This substantial reduction in both energy demand and carbon footprint makes electrochemical deposition a compelling candidate for sustainable silicon production.

The development of electrochemical silicon deposition has a long and complex history, beginning in 1865 with F. Ullik’s early experiments using K_2_SiF_6_ and KF. Throughout the late 19th and 20th centuries, researchers explored various systems—ranging from alkali metal silicates to organic solvents—yet many early efforts were limited by technical challenges, such as unsuitable electrode materials and ambiguous reaction mechanisms. Notable progress was made in the 1970s and 1980s with the introduction of fluoride-based molten salts like KF/LiF and FLiNaK, which enabled higher current densities and improved deposition efficiency. During this period, silver emerged as a promising cathode material due to its low contamination of the silicon product, outperforming graphite and platinum in terms of deposition thickness and quality. By the 1990s, research began shifting toward thin film technologies and nanoparticle synthesis, while also developing more economical methods to produce K_2_SiF_6_ from industrial byproducts. Over the past 150 years, studies have demonstrated that silicon morphology, composition, and deposition rates are highly sensitive to parameters such as electrolyte composition, temperature, electrode material, and current density. In recent decades, the focus of this research has increasingly aligned with the global demand for low-emission technologies, positioning electrochemical silicon deposition as a key candidate for sustainable innovation [2,3].

Despite the large number of studies on the electrodeposition of silicon, the question of selecting suitable technologies for large-scale industrial implementation remains open. Of particular interest is the identification of electrolyte systems that could not only partially replace the conventional metallurgical method but also potentially enable the direct production of high-purity silicon, bypassing the energy-intensive Siemens process. This work focuses on analyzing existing electrolyte systems reported in the literature for the electrochemical deposition of silicon. In this context, the morphology of the deposited material and residual impurity levels are considered secondary, as the primary objective is the production of elemental silicon suitable for further purification and industrial use.

## 2. Materials and Methods

A structured literature search was performed to identify studies on electrochemical silicon deposition in molten salts (fluorides, chlorides, mixed halides), oxide-additive systems, ionic liquids, and organic solvents. Publications were retrieved from Scopus, Web of Science Core Collection, and Google Scholar, while relevant patents (including historical chloride/fluoride systems) were sourced from Espacenet and the USPTO. The search period covered the historical record from the late 19th century through July 2025, supplemented by backward and forward citation tracking to capture seminal earlier works.

Inclusion criteria encompassed peer-reviewed articles, conference proceedings, theses, and patents that reported electrochemical deposition or electro-deoxidation routes yielding elemental silicon or Si-containing deposits, and provided experimental or mechanistic data relevant to electrolyte composition or performance (e.g., operating temperature, current density, stability, precursor solubility, morphology, current efficiency). Computational and analytical studies were included when directly addressing the silicon reduction mechanism or species behavior in the electrolytes.

For each eligible study, the following data were extracted: electrolyte composition, temperature range, silicon precursor, reported current densities and efficiencies, SiO_2_ solubility (when available), deposit morphology and purity, and operational challenges (e.g., anodic gas evolution, alloy formation). Where reported data were heterogeneous, representative ranges were compiled with contextual details (substrate, temperature, composition) to avoid over-generalization.

## 3. Results

The study of silicon electrodeposition has been ongoing for a considerable time. Among the key types of electrolytes used in such systems are solutions based on fluorides, chlorides, and their mixtures, as well as oxides, ionic liquids, and organic solvents.

### 3.1. Fluorides

As previously mentioned, the first attempts to investigate silicon electrodeposition date back to the mid-19th century. However, it was not until 1976 that Cohen and Huggins resumed these studies, utilizing a LiF-KF mixture in a 1:1 ratio with the addition of 5 mol.% K_2_SiF_6_ as a precursor. At a current density of 1–6 mA/cm^2^, a silicon layer with a thickness of 10 μm was obtained on a molybdenum cathode, forming structures of the hillock and layer types [4,5,6]. Later, G. Rao and D. Elwell investigated the process of silicon electrodeposition in the LiF-NaF-KF (FLiNaK) system, as well as in a LiF-KF (1:1) mixture at a temperature of 745–750 °C. Experiments were conducted on various cathodes using a platinum anode and a reference electrode. The K_2_SiF_6_ concentration ranged from 0 to 2.5 mol.%, and the current density varied from 3 to 60 mA/cm^2^. At a potential of −0.75 V, high-purity silicon (99–99.5%) with a thickness of 3 mm was obtained after 48 h of operation with an efficiency of 35–70% [7,8]. In subsequent experiments, the researchers increased the K_2_SiF_6_ content to 12 mol.% and applied a current density of 25 mA/cm^2^ while gradually reducing the potential from −0.8 to −0.65 V. This achieved an efficiency of 20%, and the thickness of the deposited silicon showed a linear dependence on time [9]. While studying the anodic process of silicon electrodeposition, D. Elwell and G. Rao observed a sharp increase in voltage and a transition from CO formation to CF_4_ production. This phenomenon is likely attributed to the moisture content in the melt due to the system’s high hygroscopicity [10]. Nevertheless, subsequent studies have not reported observing this effect.

In these studies, initial attempts were made to describe the mechanism of silicon reduction. G. Rao and D. Elwell proposed an explanation for the heterogeneity of the deposited silicon layer and its dissolution in the melt without the application of potential through a secondary disproportionation reaction involving the formation of the Si^2+^ ion. However, precise confirmation of this reaction became possible only in later studies conducted by Y. Suzuki and S. Nozaki [11]. Rao and Elwell also demonstrated that short electrolysis durations (with a total charge of 1200 C/cm^2^) result in low current efficiencies. This is likely due to insufficient formation of divalent silicon ions (Si^2+^), which prevents the equilibrium of the reaction from shifting toward silicon formation [12]. It is assumed that increasing the electrolysis time allows equilibrium to be reached, forming an excess of Si^2+^, which promotes further silicon reduction. Additionally, a hypothesis was proposed regarding the existence of polynuclear complexes of the Si_n_F_2n+2_ type, whose slow dissociation precedes the charge transfer stage [7,8].

K. Carleton et al., while studying the KF:LiF:K_2_SiF_6_ system (46 mol.%:46 mol.%:8 mol.%) at 750 °C and a current density of 70 mA/cm^2^ on a graphite cathode, encountered deviations from a simple diffusion-controlled model and an incomplete understanding of the process mechanism. To explain the observed phenomena, they proposed a complex mechanism involving the simultaneous occurrence of multiple reactions [13]. Meanwhile, R. Boen and J. Bouteillon, using Na_2_SiF_6_ in their experiments, hypothesized that hexafluorosilicate decomposes to form tetrafluorosilane, which then undergoes a disproportionation reaction with silicon to produce SiF_2_. Cyclic voltammetry demonstrated that the process involves two stages, each accompanied by the transfer of two electrons [14]. Further investigation into the mechanism was conducted by J. de Lepiney et al., who, using computational modeling, determined that the transition of Si^4+^ to Si^2+^ is reversible, whereas the transition of Si^2+^ to Si^0^ is characterized as quasi-reversible [15].

In many studies, silver is mentioned as a preferred material for cathodes due to its high electrical conductivity and the absence of silicon diffusion into the material. However, one of the main limitations is the low melting point of silver (961 °C) and the eutectic temperature of the Ag-Si system (845 °C) [4,5,6,7,8]. Platinum and gold have also been investigated as cathode materials. Platinum exhibits high chemical stability in halide melts; however, silicon readily forms platinum silicides through solid-state reactions at approximately 200–600 °C, and at the 600–850 °C typical of molten halide baths, extensive silicide formation and Si diffusion into platinum are expected. Gold is even less suitable at high temperatures, as the Au–Si system has a low-melting eutectic at 363 °C (~19 at.% Si), leading to the formation of a liquid phase that dissolves silicon and promotes alloying.

Despite these limitations, research conducted by A. Bieber et al. demonstrated that the rate constant for SiF_4_(g) release on a silver cathode depends on the cation of the fluoride salt. The researchers studied the fluorine acidity in mixtures of LiF–KF (51–49 mol.%), LiF–NaF (60–40 mol.%), NaF–KF (40–60 mol.%), NaF–MgF_2_ (78–22 mol.%), NaF–CaF_2_ (69–31 mol.%), and LiF–CaF_2_ (80–20 mol.%) with the addition of K_2_SiF_6_. It was found that increasing the KF content in the mixture decreases the rate constant, indicating an increase in the basicity of the environment. The most acidic composition was LiF–CaF_2_, while NaF–KF exhibited the highest basicity [16].

In a study of the KF/LiF (1:1) mixture with 10 wt% K_2_SiF_6_ in the temperature range of 750–850 °C, it was found that the use of a silver cathode resulted in higher impurity levels (approximately 100 ppm). The best results in this system were achieved using crystalline silicon (c-Si) and metallurgical silicon (MG-Si) cathodes at a current density of 10–150 mA/cm^2^ and a specific resistance of 0.013 Ω·cm [17]. On the other hand, research by K. Osen et al. showed that among cathodes made of silver, silicon, tungsten, and glassy carbon, the best performance was achieved with a silver cathode at 800 °C and a current density of 45 mA/cm^2^. However, the deposited silicon structure at 550 °C was porous, complicating further purification [18]. Haarberg, G. M. et al. continued the investigation by varying the K_2_SiF_6_ content from 5 to 20% and using different cathodes. At 800 °C, a current density of 40 mA/cm^2^, and 5 mol.% K_2_SiF_6_ on a silver cathode, the process efficiency reached 85–95%. However, experiments conducted at 550 °C with 20 mol.% K_2_SiF_6_ and a current density of 100 mA/cm^2^ led to the formation of insoluble salts on the cathode surface, significantly reducing the efficiency of the electrodeposition process [19].

The use of an iron cathode in the FLiNaK system with the addition of 6 mol.% Na_2_SiF_6_ at 750 °C, studied by Zongying Cai et al., demonstrated the formation of ferrosilicon Fe_3_Si and revealed three-dimensional nucleation of silicon. These findings indicate complex mechanisms involved in silicon formation [20]. Similarly, nickel cathodes also exhibited the formation of an alloy with silicon during electrolysis in a NaF–KF (40–60 mol.%) mixture with Na_2_SiF_6_ (0.24 mol/kg) at 850 °C and a current density of 20 mA/cm^2^. Meanwhile, on graphite and glassy carbon cathodes at high current densities (200 mA/cm^2^), silicon was deposited in dendritic structures, and a silicon carbide layer approximately 10 μm thick formed on the surface [21].

However, the formation of silicon carbide was not observed in all systems. For example, in experiments conducted by K. Stern et al. with 10 mol.% K_2_SiF_6_ in the FLiNaK system, silicon carbide (SiC) was not detected on the graphite cathode, even with the addition of 8 wt% K_2_CO_3_ [22]. Similarly, A. Abbar et al. did not report silicon carbide impurities during experiments conducted at 750 °C on a graphite cathode in the LiF/KF system (eutectic mixture) with K_2_SiF_6_ content ranging from 5 to 14%, sourced from fertilizer production facilities. The studies revealed that the peak current increased with K_2_SiF_6_ concentration up to 10%, after which it stabilized. The purity of the deposited silicon was highest at a low potential (−0.5 V on Pt). However, as the K_2_SiF_6_ concentration increased, the impurity content also rose, but silicon carbide was not among the impurities mentioned [23].

The source of silicon in the melt can be not only Na_2_SiF_6_ and K_2_SiF_6_ but also SiO_2_, which is more accessible and available in an ultrapure form on the market. The possibility of silicon electrodeposition from SiO_2_ was first investigated by D. Elwell in NaF and KF melts with the addition of BaF_2_, CaF_2_, and MgF_2_ at a temperature of 1150 °C. The maximum solubility of SiO_2_, reaching 13.8 mol.%, was observed in the CaF_2_/NaF system. Experiments were conducted on graphite electrodes at a current density of 20 mA/cm^2^. However, silicon deposition either did not occur or resulted in very small quantities. Attempts to increase the current density to 100 mA/cm^2^ also did not improve the results. It is suggested that this may be due to the rapid decomposition or volatilization of K_2_SiF_6_ and Na_2_SiF_6_ compounds formed in the melt [24].

These results, while demonstrating the potential of using SiO_2_ as a silicon source, highlighted the need for further research to improve the process. In the 21st century, this topic received renewed attention when Y. Hu et al. studied the BaF_2_-CaF_2_ (50:50 wt%) system with the addition of SiO_2_ (1.06·10^−4^ mol/cm^3^) on molybdenum cathodes at 1300 °C. It was found that the process occurs in a single stage and is irreversible, with mass transfer controlled by diffusion. At a current density of 118 mA/cm^2^, the formation of a Mo-Si intermetallic compound was observed [25]. Further development of the silicon electrodeposition method from SiO_2_ was proposed by Y. Sakanaka and T. Goto, who applied electrolysis in the FLiNaK system with the addition of 0.2 mol.% SiO_2_ at 600 °C on a silver cathode. The authors suggested that SiO_2_ dissolves in the melt, forming fluorides, oxyfluorides, and silicon oxide ions that coordinate into various forms. As a result, current peaks for each type of ion overlapped, creating a broad region. The maximum current density reached 12 mA/cm^2^. This study also introduced the use of an inert hercynite anode [26].

The exploration of alternative approaches to improve the electrodeposition process has also included the combination of SiO_2_ and K_2_SiF_6_. O.I. Boiko and Yu. K. Delimarskii were the first to propose using a combination of K_2_SiF_6_ and SiO_2_ as electrolyte components. They developed a composition consisting of 18.1% NaF, 42.4% KF, 32.5% K_2_SiF_6_, and 7.0% SiO_2_. Experiments were conducted on molybdenum, tungsten, nickel, and copper cathodes with a graphite anode in the temperature range of 710–780 °C and at current densities of 1.6–2 A/cm^2^. However, the cathodic product was heavily contaminated, with silicon content ranging from 25 to 35% at 710 °C and decreasing to 10–15% at 780 °C [27]. Building on these findings, Oishi, T. et al. proposed a method for silicon electrodeposition on a liquid Al-Si cathode at 1000 °C in a NaF-AlF_3_-SiO_2_ system to enhance process characteristics at temperatures above silicon’s melting point. In their experiments, 27 g of quartz tubes were added to 150 g of cryolite, using an aluminum cathode and a graphite anode. This approach yielded a silicon–aluminum alloy with a silicon content of approximately 50% at a current density of about 22 mA/cm^2^. The silicon was purified by washing with NaOH. While the exact current efficiency was difficult to evaluate, the authors estimated it to be 46% [28]. The various fluoride-based systems explored for silicon electrodeposition, along with their typical process parameters and outcomes, are summarized in Figure 1.

### 3.2. Chlorides

Chloride mixtures were proposed as an alternative to fluoride ones due to the aggressive environment, high melting point of fluorides, and their poor solubility in water. The first attempts to use chloride mixtures instead of fluoride ones date back to 1959, when D. Stern et al. patented a process for producing silicon in an electrolyte consisting of 61 wt% KCl, 39 wt% NaCl, and 15–20 wt% K_2_SiF_6_ at temperatures ranging from 760 to 1000 °C. The experiments involved using an iron cathode and a silicon carbide (SiC) anode. The resulting product was silicon with a purity of 92–99%. However, little is known about the details of these studies [29].

In the analysis of mixtures consisting of 100 wt% KCl, 38 wt% KCl–59 wt% CsCl, 30.3 wt% LiCl–15.3 wt% KCl–54.4 wt% CsCl, and 8.7 wt% LiCl–9.8 wt% KCl–81.5 wt% CsCl, each containing 3–5 wt% K_2_SiF_6_, it was found that the maximum current density (85 mA/cm^2^) was achieved in the KCl–CsCl system at 700 °C. However, the system with a composition of 8.7 wt% LiCl–9.8 wt% KCl–81.5 wt% CsCl also demonstrated good results (58 mA/cm^2^), despite the melting temperature of this melt being 150 °C lower. Ternary systems of LiCl–KCl–CsCl with the addition of K_2_SiF_6_ exhibited the highest rate of silicon electrodeposition. It was established that reducing the LiCl content slows down the chemical reaction responsible for the formation of volatile compounds. The morphology of the deposit strongly depends on the melt composition: in systems based on KCl and KCl–CsCl, silicon deposits in the form of fibers, whereas in melts containing LiCl, either continuous films up to 1 μm thick or dendritic structures are formed. The addition of CsCl reduces the melting temperature to 690 °C and decreases the average fiber diameter. Furthermore, an increase in overpotential enables the production of thinner and longer fibers. The addition of 0.34 wt% SiO_2_ to the melt improves the volumetric uniformity of the deposit and alters its morphology, reducing the average particle size [30,31,32,33,34,35]. Additionally, the study of the process in a KCl electrolyte with the addition of 5 wt% K_2_SiF_6_ demonstrated a rapid decrease in peak current density over time. The diffusion coefficient was found to be lower compared to fluoride-based melts, and the process itself was characterized as a linear diffusion-controlled process [36].

Despite the progress in using K_2_SiF_6_ in chloride melts, the search for alternative silicon sources continued. T. Matsuda et al. proposed using SiCl_4_ in a KCl–LiCl eutectic melt. However, SiCl_4_ was found to be insoluble in this electrolyte, which limited the current density to less than 5 mA/cm^2^, and silicon deposition occurred only at the end of the electrode opposite to the SiCl_4_ injection point [37].

One significant drawback of electrolytes based on alkali metal chlorides is their inability to dissolve SiO_2_, which limits the economic feasibility of industrial implementation. As an alternative, a CaCl_2_-based electrolyte was proposed, which is not only highly soluble in water but also effectively dissolves SiO_2_, making it more promising for practical applications. The use of CaCl_2_ in electrolytes was first investigated by Japanese researchers T. Nohira and K. Yasuda. In their experiments, silicon was produced in an electrolyte based on CaCl_2_ at 850 °C and in a LiCl–KCl–CaCl_2_ mixture (LiCl:KCl:CaCl_2_ = 52.3:11.6:36.1 mol%) at 500 °C. A single-crystal SiO_2_ wrapped in molybdenum wire was used as the working electrode, while graphite served as the anode. The results demonstrated that silicon with 90% purity could be obtained at a voltage of 0.7 V relative to the Ca^2+^/Ca pair. However, the current density did not exceed 170 mA/cm^2^, and the formation of a Si–Ca alloy was unavoidable. The alloy was then treated in an aqueous NH_4_Cl solution, which dissolved calcium as CaCl_2_ and released hydrogen, leaving silicon intact [38]. Other reported methods include leaching with dilute acids, thermal oxidation followed by leaching, and electrochemical techniques such as pulse potentials or using substrates with low calcium affinity to reduce alloy formation. Subsequent studies of the reaction mechanism revealed that during the deoxidation of SiO_2_ to Si, cracks form, which are filled with the molten salt, creating a new three-phase interface between amorphous silicon (a-Si), SiO_2_, and CaCl_2_. Further reduction occurs at this interface. The mechanism of silicon formation means that it is likely that amorphous Si forms first and then transforms into crystalline Si through a bond-breaking process [39].

It was also found that the crystallization rate of amorphous Si in CaCl_2_ at 850 °C is six orders of magnitude higher (10^−6^ m/s) than at 500 °C in a LiCl–KCl–CaCl_2_ mixture (10^−12^ m/s). The lowest achievable temperature for the process is 400 °C; however, at this temperature, the conductivity of silicon is too low for it to serve as an effective electron pathway for reduction [40]. Nevertheless, the use of ultrapure quartz crystal plates in industrial processes seems impractical. To address this, S. Cho et al. demonstrated the feasibility of using SiO_2_ nanoparticles in CaCl_2_ at 850 °C instead of single crystals [41]. Later, K. Yasuda et al. used SiO_2_ granules with a diameter of 0.25–7 mm, placed on a silicon plate at the bottom of the reactor, to reduce product contamination. A maximum current of 1.4 A was achieved for particles sized 1–2 mm after 10 h, with the thickness of the granule layer having no significant effect [42]. Additionally, the use of a BaCl_2_–CaCl_2_–NaCl eutectic with 1 mol% SiO_2_ at 650 °C on a molybdenum cathode was reported to yield porous silicon with a mixed crystalline structure [43].

Hongwei Xie et al. investigated the CaCl_2_ system with the addition of 1 wt% SiO_2_, using graphite as a substrate, and demonstrated that the morphology of the deposited silicon could be effectively controlled by adjusting the voltage in the electrolytic cell. They identified a transitional SiC layer with a thickness of less than 1 μm and explored how the deposition process impacts the structural properties of silicon. Their study further revealed the potential of electrochemically deposited silicon layers, 4 μm thick, as anodes for lithium-ion batteries, achieving a remarkable specific capacity of 2950 mAh/g—approximately nine times higher than that of traditional graphite anodes (372 mAh/g) [44]. A different approach to tailoring silicon morphology in molten salt systems was presented by Wei Weng et al. They synthesized template-free silicon nanotubes in a NaCl–CaCl_2_ molten salt (50 mol–50 mol%) using a mixture of SiO_2_ and AgCl at 850 °C. Their method achieved a high current efficiency of 74%, producing silicon with a distinctive nanotube structure. The use of solid cathodes, such as SiO_2_, SiO_2_–Ag mixtures, and SiO_2_–AgCl mixtures, sandwiched between Ni or Mo meshes, further emphasized the role of experimental conditions in shaping the final silicon morphology [45].

The question of combining K_2_SiF_6_ with CaCl_2_ remains unresolved, likely due to the high melting point of calcium chloride and the low thermal decomposition temperature of the silicon precursor, which limits the system’s stability under operating conditions. Despite this, chloride electrolytes have proven effective in addressing key challenges such as reducing the corrosive activity of the medium, enhancing the solubility of salts for purification, and partially lowering the process temperature. However, significant drawbacks persist, including the evolution of gaseous chlorine at the anode and the formation of alloys during electrodeposition in CaCl_2_. Nevertheless, these limitations have not deterred researchers from exploring the use of calcium chloride in combination with calcium oxide, which will be discussed separately. The various chloride-based systems explored for silicon electrodeposition, along with their typical process parameters and outcomes, are summarized in Figure 2.

### 3.3. Mixtures of Fluorides and Chlorides

The first attempts to study silicon deposition in chloride–fluoride mixtures began in the USSR in the 1960s. Initial studies were conducted using an NaCl–KCl–NaF electrolyte with Na_2_SiF_6_ at concentrations ranging from 0.5 to 3 wt% at 700 °C on graphite and platinum cathodes. These experiments revealed the reversibility of Si^2+^ formation and showed that the addition of NaF reduced the decomposition of Na_2_SiF_6_ and the volatilization of silicon tetrafluoride [46,47]. Silicon deposition has also been achieved in a low-melting-point electrolyte composed of LiF–KCl–CsCl with K_2_SiF_6_, at temperatures ranging from 600 to 850 °C. Substrates such as graphite, silicon, tungsten, copper, nickel, and iron were used to study the morphology of the resulting silicon over a wide range of current densities, up to 300 mA/cm^2^ [48,49].

Further investigations were carried out in the KCl–KF system with K_2_SiF_6_, where the reaction mechanisms were studied using chronopotentiometry and linear sweep voltammetry. These methods, later supplemented by more advanced techniques, demonstrated a one-step process involving the four-electron reduction of silicon and a three-dimensional instantaneous nucleation with diffusion-controlled growth [50,51]. In studies of the KF–KCl–K_2_SiF_6_ (10 mol%) and KF–KCl–K_2_SiF_6_ (10 mol%)–SiO_2_ (2–3 mol%) systems with KF-to-KCl molar ratios of 2 and 0.8, conducted at 650–800 °C on a graphite electrode, the best results were obtained at a ratio of 0.8 at 700 °C. Under these conditions, a current density of 150 mA/cm^2^ produced fine-grained silicon deposits, while at a current density of 20 mA/cm^2^, silicon with a (110) growth texture was obtained. High current efficiency was observed across all conditions, ranging from 77% to 93%. The addition of SiO_2_ further enhanced the specific surface area and reduced the average grain size [52]. The morphology of silicon is highly dependent on the current density. At current densities below 100 mA/cm^2^, fiber structures were formed. In the range of 100–250 mA/cm^2^, a combination of fibrous and spongy-fibrous structures was observed. At higher current densities, the fiber structures degraded and lost their integrity [53].

A deeper investigation into the effects of oxygen-containing additives, such as KOH and SiO_2_, on the KF–KCl (molar ratio 2:1)–K_2_SiF_6_ melt at 700 °C using a glassy carbon electrode revealed that increasing the temperature led to the growth of a single cathodic peak and an increase in silicon concentration. The ratio of *i*_p_ to *v*^1/2^ was not linear, suggesting that at relatively low scan rates, the process might be diffusion-controlled. It was claimed that fluoride, oxyfluoride, and silicate silicon complexes formed only after 100 min of sustained operation. Furthermore, it was demonstrated that the dissolution of SiO_2_ occurs gradually, with the peak current increasing proportionally to the holding time. The addition of KOH reduced the peak current for silicon, likely due to its reaction with K_2_SiF_6_, forming SiO_4_^4-^ and oxyfluoride silicon groups [53]. A more detailed study of the interaction between SiO_2_ and the electrolyte revealed the gradual transformation of [SiO_4_]^0^ into [SiO_4_]^4−^, as well as the formation of various structures such as [SiO_3_F]^3−^ and [Si_4_O_7_F_2_]. It was also shown that the melting point of the KF (40.5 mol%)–KCl (49.5 mol%)–K_2_SiF_6_ (10 mol%) system is 619 °C. Weight loss in the temperature range of 720–800 °C was measured at 1.4%, primarily due to the volatilization of SiF_4_. However, this value was relatively low compared to studies of K_2_SiF_6_ alone, as the melt interacts with KF to form K_3_SiF_7_. The addition of 0.9 mol% SiO_2_ increased the weight loss to 5.9% due to the formation of volatile SiF_4_. The melt also exhibited reactions leading to the formation of potassium oxide and potassium silicate at high temperatures, with silicon present in two forms: a silicate form and an oxyfluoride form. The reduction of silicon was confirmed to occur in a single step involving the transfer of four electrons [54]. The solubility of SiO_2_ in the eutectic KF–KCl mixture was found to be highly temperature-dependent. In the KF–KCl (2:1)–10 mol% K_2_SiF_6_ system, the solubility of SiO_2_ increased from 3.38 mol% to 7.41 mol% in the temperature range of 700–775 °C. Additionally, increasing the KF content resulted in a linear increase in SiO_2_ solubility. The solubility study also revealed that the maximum SiO_2_ solubility of approximately 10 mol% was observed at 775 °C with 10 mol% K_2_SiF_6_. Notably, the SiO_2_ content did not affect the melt’s density [55].

In the study of the KF (40.5 mol%)–KCl (49.5 mol%)–K_2_SiF_6_ (10 mol%) melt, various cathode materials—graphite, silver, glassy carbon, tungsten, and nickel—were examined for silicon deposition at current densities ranging from 10 to 150 mA/cm^2^ and temperatures of 943–1103 K. The resulting silicon exhibited impurities of up to 0.1 wt% across most substrates. On graphite, silver, and glassy carbon, the deposited silicon predominantly formed a single-phase structure, whereas on nickel, both silicon and a Ni_2_Si alloy were observed. The morphology of silicon varied significantly depending on the cathode material: columnar structures were obtained on graphite, glassy carbon, and silver, while a layer-like structure formed on tungsten. Importantly, phases such as silicon carbide and the Si/Ag intermetallic compound, commonly mentioned in other studies, were not detected [56]. Beyond solid cathodes, liquid gallium has also been explored as a novel cathode material. In this setup, silicon becomes embedded within the gallium cathode, and the process transitions from being diffusion-controlled to charge transfer-controlled. However, a key limitation of using liquid gallium is the formation of a gallium–potassium alloy, which complicates the process. Despite this drawback, the current efficiency achieved in experiments with liquid gallium reached an impressive 80% [57].

Japanese researchers studied an electrolyte composed of eutectic KF–KCl (45:55 mol%) with K_2_SiF_6_ concentrations ranging from 0.50 to 5.0 mol% on an Ag substrate at 750 °C under various current densities (10–310 mA/cm^2^). It was found that at a current density of 310 mA/cm^2^, the potential shifted to a more negative region than that of potassium. At a concentration of 0.50 mol% K_2_SiF_6_, silicon deposition did not occur. A smoother silicon structure was observed at lower current densities, while at low K_2_SiF_6_ concentrations and high current densities, silicon deposition was absent altogether. At high current densities and K_2_SiF_6_ concentrations, silicon exhibited a porous, coral-like morphology [58]. Further investigation revealed that the current efficiency at a current density of 60 mA/cm^2^ was 93%, and the process proceeded in a single step involving four electrons, with a quasireversible mechanism [59]. Other studies indicated that the second peak in cyclic voltammetry, which suggests a two-step process, only appeared with increased K_2_SiF_6_ concentrations. Special attention was given to the purity of the deposited silicon: without the addition of SiO_2_, the silicon purity analyzed by EDS reached up to 100 wt%. However, trace impurities of boron and phosphorus at several ppm levels were detected, rendering the material unsuitable for solar panel applications [60]. The effect of temperature was also studied, revealing that the process was irreversible. Higher temperatures improved crystallinity with a preferred orientation toward the <110> direction. Additionally, the diffusion coefficient increased with temperature, reaching up to 10·10^−5^ cm^2^/s at elevated conditions [61].

In contrast to these findings, the study by S. Kuznetsova et al. investigated the electrolyte NaCl–KCl (1:1)–NaF (10 wt%)–K_2_SiF_6_ at 750 °C on a silver substrate and concluded that the process occurs in two stages, each involving the transfer of two electrons. The first stage, the reduction of Si^4+^ to Si^2+^, was found to be reversible but complicated by a disproportionation (DPP) reaction at low scan rates. The second stage, the reduction of Si^2+^ to elemental silicon, was characterized as quasireversible at low scan rates, transitioning to irreversible behavior at higher scan rates [62].

The effect of alkali metal cations in AF–ACl (50:50 mol%)–A_2_SiF_6_ (1.54–3.64 mol%) melts (A = Li, Na, K, Cs) was investigated on silver and molybdenum electrodes at 800 °C. The use of a molybdenum electrode was particularly important for lithium, as it prevents the formation of an alloy. For lithium, the cathodic current for silicon overlapped with the cathodic peak corresponding to alloy formation. Studies on the decomposition of A_2_SiF_6_ revealed that in KF–KCl and CsF–CsCl, the current density remained stable over time. This stability was attributed to the melting points of Li_2_SiF_6_, Na_2_SiF_6_, K_2_SiF_6_, and Cs_2_SiF_6_, which were 613, 893, 1073, and 1033 K, respectively, indicating that the melt could not fully suppress their decomposition. During silicon deposition at current densities of 50, 100, and 200 mA/cm^2^, systems with potassium and cesium produced compact and smooth films at all current densities. In contrast, in the sodium system, compact films were obtained only at 50 mA/cm^2^, while bulky and brittle deposits formed at higher current densities [63].

The use of potassium iodide (KI) in KF–KCl–K_2_SiF_6_ melts has been explored for silicon deposition. In the KF–KCl (2:1)–75 mol% KI–0.5 mol% K_2_SiF_6_ system on a glassy carbon electrode, mixed diffusion and charge transfer limitations were identified, with iodide ions strengthening the Si–F bond. High KI concentrations promoted the formation of smooth silicon thin films by increasing the melt’s electrical resistance, reducing the height of cathodic peaks, and enlarging silicon clusters (1–3 μm) while lowering their quantity. Despite reduced conductivity, the electrolyte’s properties facilitated surface smoothing of the deposited films [64,65].

Research into doping silicon in molten salts has revealed effective methods for enhancing its properties. For instance, the addition of 0.002–0.035 wt% metallic tin in a KF–KCl melt with 1 mol% K_2_SiF_6_ at 650 °C improved silicon film quality, achieving a thickness of 60 μm, current efficiencies of 85–92%, and n-type silicon with 44% of the net photocurrent of commercial silicon [66]. Neutron transmutation doping (NTD) has also been employed to convert silicon isotopes into phosphorus, creating an effective n-type dopant [67]. For p-type semiconductors, simultaneous electrochemical deposition of silicon and aluminum in a KF–KCl–KI–K_2_SiF_6_–AlF_3_ system demonstrated similar reduction potentials for silicon and aluminum, yielding samples with 87% silicon and 0.4% aluminum at 100 mA/cm^2^. These advancements highlight the potential of molten salt systems for tailoring silicon for specific semiconductor applications [68].

Thus, fluoride–chloride mixtures successfully combine the advantages of both salts, creating favorable conditions for industrial applications. The high current density and silicon growth rate achieved in these systems make them particularly promising. For this reason, many research groups worldwide have focused their attention on studying the process of silicon electrodeposition in fluoride–chloride mixtures. The various fluoride–chloride-based systems explored for silicon electrodeposition, along with their typical process parameters and outcomes, are summarized in Figure 3.

### 3.4. Oxides

The use of an electrolyte based solely on oxides is impractical due to their high melting points. However, there have been studies on the use of oxides as additives to modify the characteristics of the silicon electrodeposition process. Early investigations into the influence of oxides were conducted in fluoride melts, where the addition of alkali and alkaline–earth metal oxides was proposed to improve the dissolution of SiO_2_. At high temperatures, these oxides form silicates, which can participate in electrochemical reactions. R.C. De Mattei, D. Elwell, and R.S. Feigelson studied the production of silicon at temperatures above its melting point. In systems with the addition of Rb_2_O, CaO, and K_2_O, silicon deposition was not observed. In contrast, the use of lithium oxide resulted in small amounts of silicon deposition. The most successful experiments, however, were conducted in a system containing 63.2 mol% SiO_2_, 22.2 mol% BaCO_3_, and 14.5 mol% BaF_2_ at a temperature of 1370 °C and current densities ranging from 60 to 500 mA/cm^2^. These conditions yielded silicon with a purity of 99.97% and an efficiency of 40%. In these experiments, carbonates were used as oxide sources, which decompose at high temperatures [69].

Yuta Suzuki et al. investigated the coordination structure of silicon in the KF-SiO_2_ system (50:50 mol.%) at 720 °C. Their study demonstrated that SiO_2_ fully dissolves in the melt, forming ions such as [SiO_4_]^4−^, [Si_2_O_7_]^6−^, [SiO_3_F]^3−^, [Si_2_O_5_]^2−^, and [SiO_3_]^2−^. Similar results were observed in the FLiNaK system containing 5 mol.% SiO_2_, which has a lower melting point and viscosity. The addition of 3 mol.% Li_2_O further increased the concentration of [SiO_3_F]^3−^ ions. In laboratory experiments on silicon deposition from the FLiNaK system with 5 mol.% Li_2_O and 5 mol.% SiO_2_ at 600 °C using a silver cathode, the addition of lithium oxide was found to significantly enhance the current density. Current efficiency increased from 13% to 50%, and the thickness of the deposited silicon layer grew from 1 μm to 100 μm [70].

Significant research has been conducted on calcium chloride-based systems, particularly mixtures of CaCl_2_ with calcium oxide and silicon dioxide precursors. Early work by O.E. Kongstein et al. examined a mixture of 81 mol% CaCl_2_, 5 mol% NaCl, 10 mol% CaO, and 4 mol% SiO_2_ at 850 °C on molybdenum and tungsten cathodes, highlighting challenges with silicon anode passivation and achieving a current density of 150 mA/cm^2^ using square current pulses [71]. Later studies demonstrated advancements in producing photoactive silicon films for solar panels. Ji Zhao et al. obtained 3.5 μm thick films in a CaCl_2_ melt with 1.81 mol% SiO_2_ nanoparticles at 6 mA/cm^2^, achieving 31% photocurrent compared to standard silicon wafers [72]. Xiao Yang et al. improved this process, achieving 20 μm thick films with 40% current efficiency and 36% photocurrent in a CaCl_2_–CaO–SiO_2_ melt at 850 °C and 15 mA/cm^2^ [73]. Xingli Zou et al. further developed the approach to produce p-type, n-type, and p-n junction silicon films by adding alumina, boric anhydride, or calcium phosphate, achieving photocurrents of 40–50%. Pre-electrolysis for 120 h ensured high gas purity, with 97% of the evolved gas being carbon monoxide [74]. Zou also synthesized Si, Ge, and SiGe micro/nanowires in a CaCl_2_ melt with 1.5–5 wt% CaO and 1–3 wt% SiO_2_ or GeO_2_, finding that potential strongly influenced the morphology [75].

In their investigation of the reaction mechanisms of dissolution and electrodeposition, X. Li et al. studied the CaCl_2_–CaO (1.68 wt%)–SiO_2_ (1.8 wt%) system. They found that at temperatures above 450 °C, Ca_4_OCl_6_ begins to form as a composite reaction product of CaO and CaCl_2_. At 650 °C, this phase disappears, giving way to the formation of CaSiO_3_ and Ca_2_SiO_3_Cl_2_. The study also identified various silicon ions in the melt, including SiO_4_^4−^, Si_2_O_7_^6−^, SiO_3_^2−^, and Si_2_O_5_^2−^. The reduction of SiO_3_^2−^ was found to be diffusion-controlled, with a diffusion coefficient of 3.31·10^−5^ cm^2^/s. When the SiO_2_ content was halved, SiO_4_^4−^ became the dominant ion, and its reduction proceeded in two stages: Si^4+^ –> Si^2+^ –> Si. The study also highlighted the significant influence of current density on silicon morphology: at densities below 10 mA/cm^2^, silicon formed nanowires; from 15 to 25 mA/cm^2^, dense films were produced; and at higher current densities, irregular silicon particles were observed. In the growth of silicon films deposited on silicon substrates with orientations (110) and (111), distinct morphologies were noted. On (110)-oriented substrates, inverted pyramidal structures formed, whereas (111)-oriented substrates produced dense triangular prismatic structures. Additionally, the resistivity of the deposited silicon was comparable to that of the original substrates [76].

Dong, Y. et al. investigated the use of CaSiO_3_ in a calcium chloride-based electrolyte. The addition of CaO was studied to enhance reaction kinetics limited by the diffusion of O^2-^ ions. Although CaO did not directly participate in the reaction, it was hypothesized that a higher concentration of CaO could increase the ionic flux of O^2-^ ions in the melt, thereby improving reaction kinetics. Adding 1.25 wt% CaO to the CaCl_2_ melt increased the silicon yield nearly threefold and doubled the current density. However, further increasing the CaO content did not result in significant improvements, even though the concentration remained well below the solubility limit of CaO in CaCl_2_. To reduce the melting temperature, a eutectic CaCl_2_–NaCl system with a melting point of 601 °C was proposed. However, the use of NaCl reduced the silicon yield threefold, from 55 mg to lower levels, and the solubility of CaSiO_3_ in this system was only 0.2 wt%. Replacing NaCl with MgCl_2_ slightly improved the silicon yield, lowered the process temperature to 700 °C, and increased the solubility of CaSiO_3_ to 0.8 wt%. The most optimal composition, with a NaCl/CaCl_2_/MgCl_2_ mass ratio of 2:4:1, achieved a CaSiO_3_ solubility of 1.0 wt% at 650 °C and 1.2 wt% at 750 °C. In this mixture, the silicon yield at 650 °C was 75 mg with ~70% Coulombic efficiency. Additionally, the authors demonstrated the possibility of using Na_2_O instead of CaO and proposed recycled glass as a substitute for CaSiO_3_ [77]. Xingli Zou et al. further showed that this electrolyte could be used to directly obtain p–n junctions with a film thickness of approximately 40 μm, low impurity content, and 60% faradaic efficiency [78].

The addition of oxides in fluoride–chloride mixtures has not been extensively studied. One investigation focused on the kinetics of an electrolyte composed of CaCl_2_–CaF_2_ (80:20 wt%)–CaO (4 mol%)–SiO_2_ (4 mol%) on a tungsten electrode at 750 °C. The process was found to be quasi-reversible and controlled by mass transfer within the applied scan rates, with a diffusion coefficient of 3.22·10^–5^ cm^2^/s. The study also indicated a single-step reaction mechanism [79]. In another study using a molybdenum electrode, the process was reversible and resulted in the formation of a MoSi_2_ alloy. The current density in this system reached 60 mA/cm^2^ [80].

Thus, oxides primarily interact with silicon precursors to accelerate their dissolution and facilitate silicate formation. However, the underlying mechanism remains incompletely understood, particularly given that alkali metal oxides have not led to silicon formation. Improvements have been observed only in systems containing calcium and barium salts. The various systems with oxide additions explored for silicon electrodeposition, along with their typical process parameters and outcomes, are summarized in Figure 4.

### 3.5. Ionic Liquids

Electrochemical silicon deposition in ionic liquids is a relatively new but rapidly developing field, with initial studies conducted in the 2000s. One of the key advantages of this process is the ability to operate at low, near-room temperatures. Y. Katayama et al. first synthesized 1-ethyl-3-methylimidazolium hexafluorosilicate ((EMI)_2_SiF_6_), which was then dissolved in 1-ethyl-3-methylimidazolium bis(trifluoromethylsulfonyl)imide (EMITFSI) and subjected to electrolysis at 25 °C and a current density of 150 mA/cm^2^. On a silver cathode, the process was found to be irreversible, resulting in a transparent film primarily composed of silicon oxide. According to the authors, amorphous silicon initially formed during the reaction, which subsequently reacted with water to produce the oxide. Electrolysis of ((EMI)_2_SiF_6_) alone yielded a similar result, confirming the feasibility of producing silicon-containing films from ionic liquids [81].

Other researchers utilized 1-butyl-1-methyl-pyrrolidinium bis(trifluoromethylsulfonyl)imide ([BMP]Tf_2_N) saturated with gaseous SiCl_4_ for electrolysis on a gold cathode, resulting in nanoscale silicon deposition. Graphite was unsuitable as a cathode due to its rapid degradation. The process was confirmed to be irreversible, primarily diffusion-controlled, with the coevolution of chlorine [82]. Further studies revealed that silicon film growth was limited to a few nanometers in height, complicating additional deposition [83]. This electrolyte also enabled the formation of Ge-Si alloys by saturating it with a 1:1 mixture of SiCl_4_ and GeCl_4_, resulting in pure luminescent semiconductor nanoparticles [84]. Additional experiments using SiCl_4_ and SiBr_4_ on aluminum and nickel cathodes showed that the process was not diffusion-controlled but rather adsorption-driven. On aluminum, the electroreduction of silicon was controlled by nucleation and growth of Si nuclei, likely due to the low precursor concentration. The nucleation process was three-dimensional, with instantaneous growth of Si nuclei on the Al substrate. The current density reached 0.5 mA/cm^2^, but the deposited silicon oxidized rapidly in air and exhibited cracking. On nickel, larger Si clusters formed compared to aluminum, potentially due to the presence of an Al_2_O_3_ layer that hindered the process on aluminum. In other aspects, results on both electrodes were similar, although the silicon contained significant impurities of Cl and S [85]. C. Fournier and F. Favier reported the deposition of nanowires not only of Si but also of Ti and Zn in the same electrolyte, with lengths up to 100 nm [86]. Studies also showed that increasing the SiCl_4_ concentration to 0.6 M enhanced the solution’s conductivity and reduced its viscosity, but no further effects were observed at higher concentrations. Deposited silicon layers prevented irreversible decomposition of the cation, though the silicon content in the layer was only 31%. Substrate material played a significant role: on aluminum and copper electrodes, simultaneous decomposition of the ionic liquid was observed, a phenomenon not detected on gold [87].

Further studies using 1-butyl-1-methylpyrrolidinium bis(trifluoromethanesulfonyl)imide on gold-coated silicon (100) at 50 °C resulted in the deposition of amorphous silicon with a purity of up to 99%. However, the silicon surface was contaminated with oxygen compounds. Investigations of silicon deposition at potentials below the ionic liquid decomposition potential revealed significant structural differences, with columnar structures transitioning to laminar ones. It was also shown that the growth rate increased with higher applied potentials. Raising the process temperature from room temperature to 100 °C significantly improved deposition by reducing the solution viscosity. This resulted in silicon films with thicknesses of 3–5 μm, compared to 1 μm at room temperature, although the roughness of the deposit also increased. The addition of up to 10% acetonitrile to lower viscosity was explored but had no significant impact on the process [88].

In studies on the influence of anions on silicon electrodeposition, ionic liquids with the cation 1-butyl-1-methylpyrrolidinium ([Py_1_,_4_]^+^) and three different anions—trifluoromethylsulfonate (TfO^−^), bis(trifluoromethylsulfonyl)amide (TFSA^−^), and tris(pentafluoroethyl)-trifluorophosphate (FAP^−^)—were used with SiCl_4_ as the silicon source. The results showed that the anion composition significantly affected the silicon deposition mechanism and structure. Silicon with agglomerated spherical structures was obtained using TfO^−^, very fine spherical structures with TFSA^−^, and porous structures with FAP^−^. In all cases, amorphous silicon was produced. When copper was used as the cathode at 100 °C, a Cu_3_Si alloy was formed [89].

The influence of water on silicon deposition was investigated in 1-butyl-3-methylimidazolium bis(trifluoromethylsulfonyl)imide ([BMIm]Tf_2_N) and 1-butyl-3-methylimidazolium hexafluorophosphate ([BMIm]PF_6_) on a graphite cathode, with water contents of 0.86 wt% and 0.33 wt%, respectively. Water had little effect on the silicon deposition process, except in [BMIm]PF_6_, where it significantly reduced the sticking ability of silicon on the substrate [90].

The process was also carried out in 1-butyl-3-methylimidazolium bis[(trifluoromethyl)sulfonyl]imide ([BMIm]Tf_2_N) mixed with propylene carbonate at mass ratios of 1:1, 1:2, 1:3, and 1:4 on a titanium cathode. The process showed low dependence on temperature and ionic liquid-to-solvent ratio. However, the most regular silicon structure with minimal cracking was observed at a 1:3 ratio at room temperature. The purity of the deposited silicon was 37% [91].

Studies on trimethyl-n-hexylammonium bis(trifluoromethylsulfonyl)imide (TMHATFSI) with the addition of SiCl_4_ revealed that the dissolution of the silicon precursor occurs due to van der Waals forces between the hexyl group in the trimethyl-n-hexylammonium cation (TMHA^+^) and SiCl_4_. Electrolysis on a nickel cathode produced a 250 nm thick silicon film, with no evidence of a Ni-Si alloy, likely due to the low process temperature [92]. Further studies using a gold cathode demonstrated a silicon purity of 70%, although the deposited material was heavily contaminated with the ionic liquid [93]. Mechanistic investigations showed that amorphous silicon initially forms and transitions to a crystalline structure after annealing. The current efficiency ranged from 73% to 95% across different potentials, with the reduction occurring in a single four-electron step. Annealing was also proposed as a purification step to remove high levels of organic impurities from the silicon [94]. A more detailed analysis revealed that SiCl_4_ is first reduced to Si_2_Cl_6_, which then forms a polymer-like intermediate, ultimately being deposited as silicon on the substrate [95]. This electrolyte has also been used to produce thin Si films with p-type characteristics by co-depositing with AlCl_3_. Films obtained through electrodeposition with a rest period exhibited smoother structures, while constant-potential electrodeposition resulted in more porous films. However, the current density for these processes remained low, at approximately 0.5 mA/cm^2^ [96].

On a nickel electrode, other researchers studied the ionic liquid N-ethyl-N-methylpyrrolidinium chloride–zinc chloride at 150 °C with SiCl_4_. However, the solubility of SiCl_4_ in this system was low, and during electrolysis, zinc was first reduced at the cathode. The reduced zinc then chemically reduced silicon in parallel with the primary electrochemical process. The current density reached 0.17 mA/cm^2^ [97].

Another ionic liquid, tri-1-butylmethylammonium bis(trifluoromethylsulfonyl)amide ([N_4441_][TFSI]), which exhibited high electrochemical stability, was used with SiCl_4_ on a liquid gallium electrode at 100 °C. The deposited silicon showed an amorphous morphology on the metal surface, while within the gallium, it exhibited a crystalline diamond cubic lattice, growing from the amorphous layer. The deposition mechanism likely involved the formation of amorphous silicon, which dissolved in gallium and began crystallizing after saturation, induced by the external amorphous silicon layer [98]. When the electrolyte was changed to butyltrimethylammonium bis(trifluoromethylsulfonyl)imide ([N_1114_][TFSI]), containing a less bulky cation, silicon deposition on the same electrode was found to be diffusion-controlled, proceeding through a single step with a current density below 5 mA/cm^2^. Crystalline silicon began forming at temperatures between 80 and 120 °C. Using an indium–gallium substrate intensified the oxidation of silicon, highlighting the impact of substrate composition on the process [99].

In general, ionic liquids represent a novel class of electrolytes for silicon deposition, offering the significant advantage of efficient operation at temperatures below 100 °C. However, their major drawbacks include low current densities, the necessity of using SiCl_4_ with limited solubility, high cost, and decomposition during the electrochemical process. The various systems employing ionic liquids for silicon electrodeposition, together with their typical process parameters and outcomes, are summarized in Figure 5.

### 3.6. Organic Solvents

The previously discussed ionic liquids emerged as an alternative to earlier proposed organic solvents. The first attempts to use aprotic organic solvents for silicon deposition date back to the 1970s. A 1975 patent by A.E. Austin proposed the use of propylene carbonate, tetrahydrofuran (THF), acetonitrile, and dimethylsulfite as solvents, with SiHCl_3_, SiCl_4_, SiBr_4_, and SiI_4_ as silicon precursors. Silicon was deposited on a platinum electrode at 25–85 °C and current densities up to 2.5 mA/cm^2^, although for many solvents, the current density was limited to 0.1 mA/cm^2^. The resulting silicon exhibited high purity, with impurity levels as low as 10 ppm. The main impurities were SiO_2_ and SiO. Austin also noted the limited solubility of silicon precursors in organic solvents—an exception being THF—and the low conductivity of these systems [100].

Studies on silicon deposition in organic solvents have explored various systems, with a focus on propylene carbonate containing SiCl_4_ and tetrabutylammonium chloride (TBACl). This electrolyte has been tested on a range of substrates, including Ti-6Al-4V alloys, nickel, titanium–nickel–niobium alloys, and copper, under different conditions to investigate deposition mechanisms, product morphology, and practical applications. On Ti-6Al-4V substrates, amorphous silicon films with a thickness of 1 μm were deposited after 1000 min at 35 °C. Higher precursor concentrations (up to 1 M) and elevated temperatures improved the uniformity and smoothness of the surface but required increased pressure due to the evaporation of SiCl_4_. The films contained significant hydrogen impurities (up to 35% at 35 °C), primarily in the form of SiH_2_, which decreased at higher temperatures [101]. On nickel substrates, silicon exhibited a columnar morphology with a current density of 3.4 mA/cm^2^. However, the process was complicated by the reduction of TBA^+^, leading to gas evolution and the formation of a gel-like amorphous silicon film [102]. Similar issues were observed on titanium–nickel–niobium alloy electrodes, where the process was found to be irreversible, yielding silicon with 33% purity. The material was highly reactive and oxidized rapidly upon exposure to air [103]. Copper substrates produced primarily amorphous silicon with some nanocrystalline inclusions. At a current density of 1 mA/cm^2^, the deposited films demonstrated promising performance as anodes for lithium-ion batteries, achieving a reversible capacity of ~1300 mAh/g compared to the theoretical 4200 mAh/g for silicon [104]. J. Gu et al. achieved fully crystalline black silicon with grain sizes of ~500 nm and a purity of 99 at% using a liquid gallium electrode at 100 °C [105]. Photoactive silicon films were deposited on gold electrodes in a propylene carbonate solution of SiHCl_3_ and tetrabutylammonium bromide (TBAB). High oxygen concentrations (29–39 at%) were observed in the films, likely due to insufficient reduction potential or solvent decomposition. The films exhibited photocurrent at potentials from −2.5 V to −2.7 V vs. Ag but not at −2.85 V [106]. Pulsed current electrodeposition in propylene carbonate with 0.1 M TBACl and 0.5 M SiCl_4_ on copper foils produced amorphous yellow silicon. Increasing the pulse frequency from 0 to 5000 Hz reduced particle size and transitioned the morphology from islands to thin films, likely due to changes in the growth mechanism [107].

Tetrahydrofuran (THF) has been investigated as an electrolyte due to its excellent solubility for silicon halides and related precursors. Early studies explored the use of SiHCl_3_, SiCl_4_, SiBr_4_, Si(CH_2_CH_3_)_4_, Si(OCH_2_CH_3_)_4_, Si(OOCCH_3_)_4_, and Si[N(CH_3_)_2_]_4_ in THF with supporting electrolytes such as LiClO_4_, tetrabutylammonium bromide (TBAB), and tetrabutylammonium perchlorate (TBAP). The results revealed that SiI_4_ and SiBr_4_ were less chemically stable, with irreversible reduction processes. No significant electrochemical activity was detected for organosilicon compounds, leading researchers to conclude that SiHCl_3_, SiCl_4_, and SiBr_4_ were the most suitable precursors. However, the maximum current density across all systems was limited to 1.4 mA/cm^2^, attributed to potential cathode passivation and high film resistivity. The current efficiency for SiHCl_3_ was 60%, while for SiCl_4_ it was 35%, yielding silicon films 0.25 μm thick with a silicon purity of 82% [108]. Further investigations using THF and acetonitrile with 0.3 M SiCl_4_ or 0.3 M SiHCl_3_ revealed significant effects of substrate material on morphology. Deposits on glassy carbon were nodular with few cracks, while those on Ni formed agglomerates, and those on Pt exhibited non-uniform plate-like structures. Films derived from SiHCl_3_ oxidized more slowly than those from SiCl_4_ due to hydrogen termination of Si bonds, with local silicon purity reaching 91%. Increasing the deposition temperature to 70 °C did not improve film quality, though metals like Sn or Pb accelerated the crystallization of amorphous silicon [109]. In another study, deposition in THF with 0.3 M SiCl_4_ and 0.1 M tetrabutylammonium chloride (TBAC) on TCO (aluminoborosilicate glass/SnO_2_:F) cathodes produced silicon films composed of 78 at% Si, 12 at% O, 6 at% C, and 4 at% Cl. Pre-drying the organic solvent reduced oxygen content in the final product to 9%, suggesting that oxygen impurities arose from moisture in the precursor or supporting electrolyte. The current efficiency reached 28%. Post-deposition annealing above 400 °C was necessary to stabilize the silicon against oxidation, while annealing at 800 °C resulted in the formation of Si crystallites less than 10 nm in size within an amorphous matrix [110].

Beyond the commonly studied propylene carbonate and tetrahydrofuran, additional research has explored a variety of solvents and supporting electrolytes for silicon deposition, aiming to expand the scope of suitable systems. J. P. Nicholson investigated silicon deposition using various supporting electrolytes, such as tetraethylammonium chloride (TEAC), tetrabutylammonium chloride (TBAC), and lithium chloride (LiCl), alongside solvents like propylene carbonate (PC), tetrahydrofuran (THF), and acetonitrile (AN), with SiCl_4_ and SiBr_4_ as precursors. Experiments on titanium substrates and n-type silicon wafers revealed declining efficiency over time due to increasing layer impedance. Lithium chloride proved ineffective due to low solubility, and deposited silicon oxidized rapidly at current densities of 0.4–1 mA/cm^2^. Better results were achieved on n-type silicon wafers, where impurities penetrated up to 200 nm before forming a purer silicon layer. THF demonstrated a higher reduction potential than PC, but no significant differences were observed between SiCl_4_ and SiBr_4_ precursors. The potential formation of Si_2_Cl_6_ as an intermediate was also suggested [111].

M. Bechelany et al. investigated silicon deposition from acetonitrile (CH_3_CN) and dichloromethane (CH_2_Cl_2_) containing 1 M tetraethylammonium chloride and 1 M SiCl_4_ on gold electrodes, focusing on the oxidation process and methods to enhance product properties. Heat treatment under hydrogen in an argon vacuum was shown to significantly improve the silicon quality. In CH_3_CN, silicon deposition at low potentials was accompanied by a competing gas evolution reaction, resulting in porous, rose-like structures and a reduced film thickness of ~2500 nm. In CH_2_Cl_2_, the deposition rate was higher, with film thickness doubling compared to CH_3_CN, but the resulting silicon structure was even more porous. Gas analysis revealed the evolved gas consisted primarily of methane, ethylene, and other hydrocarbons. Silicon films from CH_3_CN contained 57 at% Si and 4 at% C, while films from CH_2_Cl_2_ contained 22 at% Si and 28 at% C, with additional impurities of O and N. Notably, oxygen impurities were surface-bound and could be removed via HF treatment [112]. Further studies in acetonitrile focused on silicon deposition for lithium-ion battery applications demonstrated significant advancements. Using a solution of 0.3 M SiCl_4_ and 0.1 M tetrabutylammonium chloride on nickel electrodes, researchers achieved a reversible capacity of 2600 mAh/g, far exceeding the performance of silicon films deposited from propylene carbonate [113].

In a system comprising ethylene glycol with H_2_SiF_6_ as the silicon source, investigated on a steel cathode at room temperature, an optimal precursor concentration of 0.2 M was identified. Higher concentrations led to gas bubbling at the cathode, disrupting the deposition process. Current densities ranged from 30 to 120 mA/cm^2^, yielding amorphous silicon films with fluoride, hydrogen, and carbon inclusions. The average film thickness was 4.3 μm. The product’s color varied with H_2_SiF_6_ concentration, ranging from green and blue to dark black, with higher concentrations improving homogeneity [114]. Surface and impurity analyses revealed prominent fluoride-related inclusions, such as SiF_3_ and SiF_2_ groups, alongside CF_3_ groups, Si–OH bonds, and significant Si–C bonds primarily within the silicon matrix [115]. In another study, ethylene glycol, HF, and silicic acid were used on a nickel cathode at a current density of 50 mA/cm^2^, achieving an efficiency exceeding 100%, indicating impurity incorporation. Adding up to 10 vol% formamide increased the silicon deposition rate, with identified impurities including organosilicon compounds such as (CH_3_)_x_Si_y_N_z_. Replacing silicic acid with tetraethyl orthosilicate (TEOS) also enabled silicon deposition. The reaction mechanism likely involved the formation of SiF_4_ through the precursor’s reaction with HF, followed by its electrochemical reduction [116]. Using TEOS in acetic acid with tetramethylammonium chloride (TMAC) on nickel at ~1 mA/cm^2^ produced a blue amorphous silicon thin film. Higher temperatures and precursor concentrations reduced current efficiency, which peaked at 60%. Hydrogen evolution began at current densities above 0.4 mA/cm^2^. Adding pyridine improved conductivity but reduced efficiency at higher current densities (up to 10 mA/cm^2^). Silica gel was the main impurity across all experiments with TEOS. Notably, no silicon deposition occurred on copper, gold, or platinum cathodes under similar conditions [117].

Other researchers successfully deposited silicon from an acetone solution containing K_2_SiF_6_ with 6 vol% HF, achieving a current density of 1.5 mA/cm^2^. The resulting product did not contain hydrogen compounds. When triethylborate was added to the solution, no boron was detected in the final product. In contrast, the addition of triethylphosphite resulted in phosphorus incorporation into the silicon structure [118]. A. Abbar and S. Kareem investigated silicon deposition from acetone containing 0.6 M phenyltrichlorosilane and 0.07 M tetrabutylammonium chloride using various cathodes. The best results were achieved with titanium and amalgamated copper. However, at high precursor concentrations, significant decomposition of the organic solvent was observed. The coulombic efficiency of the process ranged between 81% and 87%, and the product primarily consisted of Si, O, and Cl. The silicon particles had an average size of 160–165 nm. On titanium, the deposited silicon exhibited finer corrugation and a more amorphous structure, while on copper, amorphous silicon could not be obtained [119].

Organic aprotic solvents offer several advantages, including relatively low cost, the ability to quickly purify the obtained product, and the possibility of conducting processes at moderate temperatures. However, these benefits are outweighed by significant drawbacks, such as decomposition under the influence of electricity, low conductivity that necessitates the use of additional supporting substances, and the predominant reliance on gaseous silicon precursors. As a result, their application remains limited. An overview of silicon electrodeposition systems using organic solvents, along with typical process parameters and results, is provided in Figure 6.

## 4. Discussion

Based on the conducted literature review and other analyses of studies [120,121], it can be concluded that the successful industrial implementation of electrochemical silicon extraction requires optimal process parameters, with a particular focus on the composition of the electrolyte. The ideal electrolyte should have high conductivity and low resistance, be chemically stable under process conditions, effectively bind silicon precursors (e.g., K_2_SiF_6_) in high concentrations, and possess the ability to dissolve SiO_2_, which is the most accessible and cost-effective raw material for silicon production. Additionally, it is important to ensure a low process temperature to reduce energy costs and achieve high product yield by minimizing irreversible chemical reactions with silicon precursors and intermediate products. The electrolyte should be readily available on the market, easily regenerable, and efficiently removable from the final product. The silicon obtained in the process should have a crystalline structure, be resistant to oxidation in air, and minimally interact with the substrate material to prevent alloy formation. Thus, selecting an electrolyte that meets these requirements is a critical factor for the industrial application of electrochemical silicon extraction technology.

Among all the considered electrolytes, fluoride-based systems are the most extensively studied and offer significant advantages that contribute to efficient process execution. In fluoride systems, silicon dioxide dissolves effectively, they exhibit high conductivity, and the highest current density values have been achieved in these systems. Their behavior in melts is well-researched, making them attractive for industrial applications. However, fluoride systems also have significant drawbacks. These include a high melting temperature, which increases energy costs, as well as high hygroscopicity, which can degrade electrolyte properties and lead to the occurrence of the anodic effect. The anodic effect, in turn, can cause a sudden increase in anode voltage, reducing process stability. Additionally, the relatively low solubility of fluoride electrolytes in water complicates product purification and reduces the economic efficiency of the process.

Chlorides offer several advantages over fluorides, including better solubility in water, which is crucial for silicon purification, the absence of HF impurities, lower overall aggressiveness, reduced hygroscopicity, and lower melting temperatures. However, the electrolysis of chlorides is accompanied by the release of Cl_2_ at the anode, which requires additional measures to manage the process. Additionally, alkali metal chlorides do not dissolve SiO_2_ and form weak coordination bonds with SiF_6_^2−^.Alkali metal chlorides exhibit relatively good current density performance, but their inability to dissolve SiO_2_ limits their applicability. On the other hand, alkaline earth metal chlorides have sufficient solubility for oxide ions, yet the current density in these systems remains low, reducing their efficiency. Chloride-based systems are actively researched for producing nanotubes and thin films of ultrapure silicon at low current densities and on various substrates. However, the exclusive use of chloride electrolytes for industrial-scale silicon production is likely impractical due to the limitations.

Mixtures of fluorides and chlorides combine the positive properties of both groups of electrolytes. They exhibit high current density, have low melting temperatures characteristic of alkali metal chlorides, effectively dissolve silicon dioxide, and, importantly, do not result in chlorine evolution at the anode. They exhibit high current density, have low melting temperatures characteristic of alkali metal chlorides, effectively dissolve silicon dioxide, and, in fluoride-rich compositions, suppress chlorine evolution at the anode by widening the anodic electrochemical window. In contrast, chloride-rich mixtures narrow this window, making Cl_2_ release possible if the anodic limit is exceeded. For example, the LiCl–KCl eutectic has an electrochemical window of approximately 3.67 V at 773 K. Despite these advantages, the melting temperatures of such systems remain relatively high, ranging from 600 to 900 °C, which may limit their applicability in certain processes.

The addition of oxides, according to some researchers, improves process performance, particularly in CaCl-CaO systems. However, studying their behavior remains a complex task, especially in systems containing K_2_SiF_6_. In electrolytes based on alkali metals, the influence of oxides is limited and has an additional impact on the process. Furthermore, the solubility of the oxide in the electrolyte must also be considered. It is likely that their reaction mechanism in the melt is closely linked to the formation of silicates from SiO_2_. In the presence of fluoride mixtures, this reaction may become competitive, potentially affecting the overall process efficiency.

Organic electrolytes, such as ionic liquids and organic aprotic solvents, compensate for the high melting temperatures characteristic of other systems due to their low-temperature nature. However, they have several significant drawbacks, including difficulties in moisture control, low electrical conductivity, high cost, and limited chemical stability under the influence of electric current. Additionally, they are incompatible with the most common and easy-to-use silicon precursors, such as K_2_SiF_6_ and SiO_2_. The morphology of silicon obtained in such systems is predominantly amorphous, which is highly prone to oxidation in air. Furthermore, the current densities achieved at this stage of research are insufficient for industrial applications. Thus, the use of organic electrolytes for silicon production is likely to remain of fundamental research interest and is not yet suitable for practical industrial implementation. A comparison of all electrolyte systems discussed is presented in Table 2.

Another important parameter of the process is the choice of substrate material. Among various substrates, silver is the most studied due to its high conductivity and weak interaction with the resulting silicon. However, silver has certain disadvantages: a relatively low melting point (961 °C) and high cost. Nickel and iron cathodes are prone to the formation of alloy layers with silicon, which can limit their application. Aluminum substrates are suitable only in their liquid form with silicon incorporation. Graphite and glassy carbon are of interest due to their electro-physical properties, but their brittleness and the potential formation of undesirable silicon carbide, which is not confirmed by all researchers, reduce their attractiveness. Molybdenum and tungsten electrodes are also widely used in research. Despite the formation of alloys on their surface, these materials exhibit high resistance to aggressive environments and enable experiments at high temperatures, making them promising candidates for use in electrochemical processes.

## 5. Conclusions

Thus, from an industrial perspective, mixed fluoride–chloride melts containing K_2_SiF_6_ and SiO_2_—particularly KF–KCl systems—currently offer the most balanced combination of high current density, effective SiO_2_ dissolution, and suppressed chlorine evolution. Tungsten and molybdenum substrates have shown the best stability under these conditions. However, existing studies are limited to specific molar ratios (1:1 and 2:1), and comprehensive data on achievable silicon purity remain scarce. Compared with organic solvents and ionic liquids, which are constrained by low current densities and limited scalability, mixed fluoride–chloride systems appear more viable for metallurgical- and polycrystalline-grade silicon production in the near term.

Several open questions remain in this field. These include the influence of temperature, substrate material, precursor concentration, and the role of chloride and fluoride cations on the kinetics of silicon deposition. While some studies have investigated different metal cations, most experiments used various unstable silicon salts, without systematically analyzing how chloride and fluoride ions affect the kinetics of silicon electrodeposition from K_2_SiF_6_.

Despite nearly 175 years of research, no consensus has been reached regarding the fundamental nature of the electrochemical process. There is ongoing debate over whether the reduction is reversible, quasi-reversible, or irreversible, and whether it occurs through a single-step or two-step mechanism. Addressing these challenges through systematic experimental work, coupled with advanced in situ diagnostics, will be crucial to optimizing process parameters, minimizing unwanted alloy formation, and identifying electrolyte systems capable of scalable, low-emission silicon production, while complementary computational modeling can provide predictive insight into thermodynamic stability, kinetic pathways, and process scalability. This combined approach will enable the identification of the most promising electrolyte compositions and operating conditions for industrial implementation.

## Figures and Tables

**Figure 1 materials-18-04009-f001:**
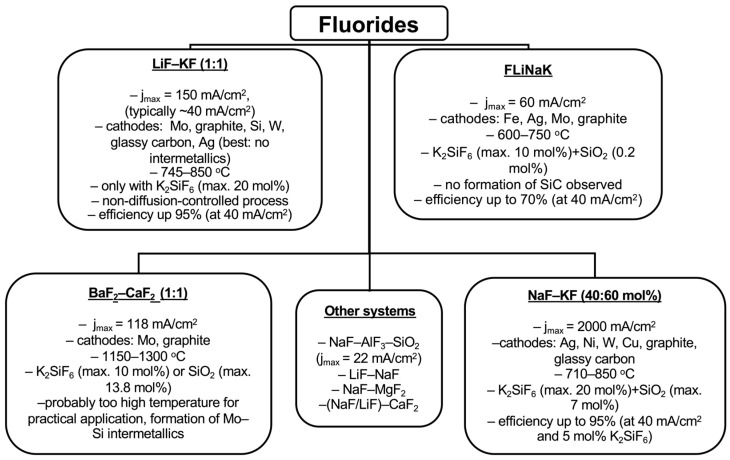
Overview of fluoride-based electrolyte systems for silicon electrodeposition.

**Figure 2 materials-18-04009-f002:**
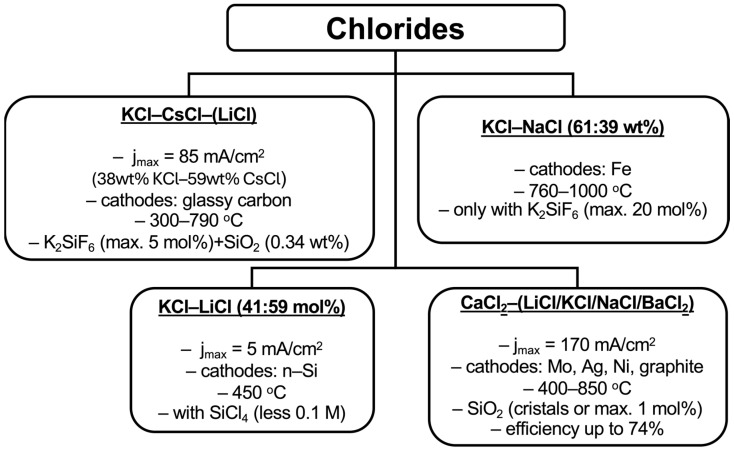
Overview of chloride-based electrolyte systems for silicon electrodeposition.

**Figure 3 materials-18-04009-f003:**
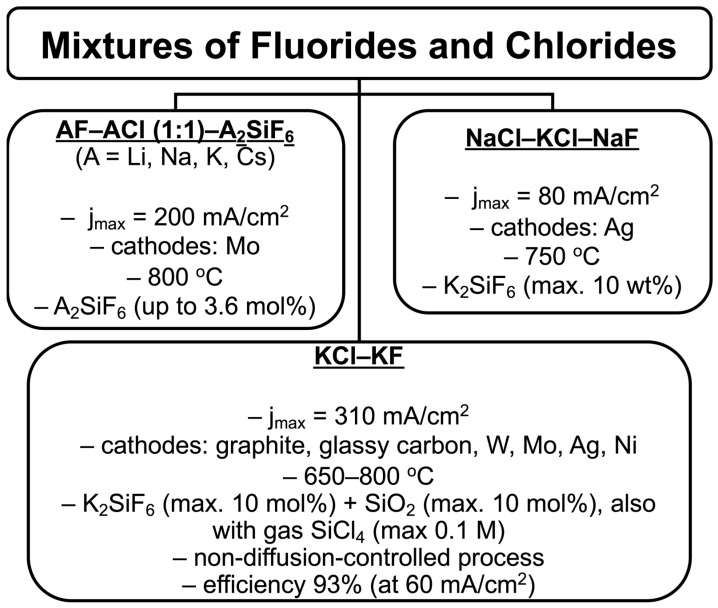
Overview of fluoride–chloride-based electrolyte systems for silicon electrodeposition.

**Figure 4 materials-18-04009-f004:**
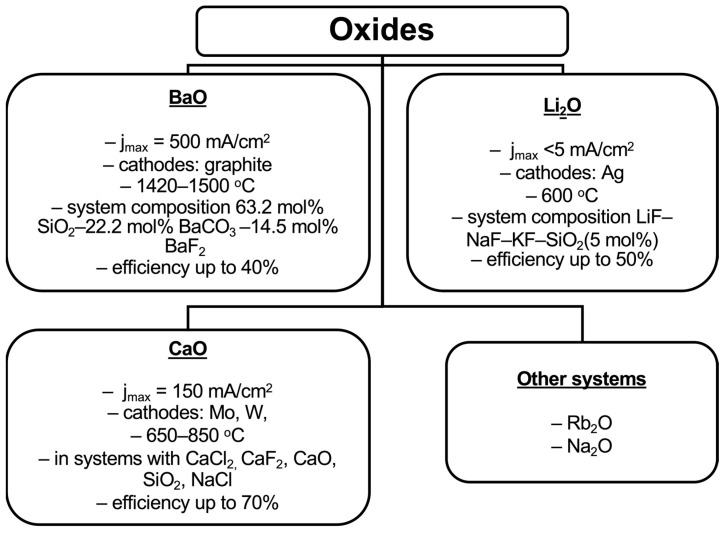
Overview of systems with oxide additions for silicon electrodeposition and their typical process parameters.

**Figure 5 materials-18-04009-f005:**
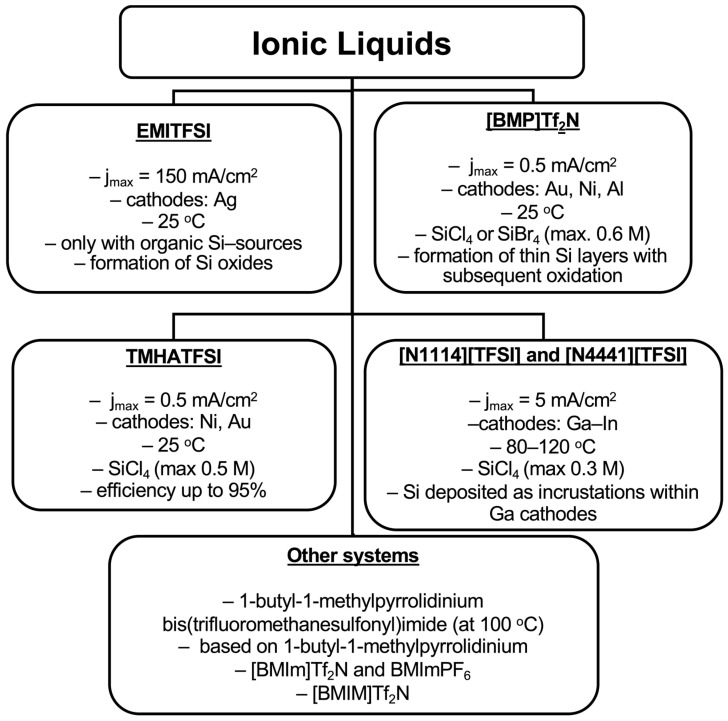
Overview of ionic liquid systems for silicon electrodeposition and their typical process parameters.

**Figure 6 materials-18-04009-f006:**
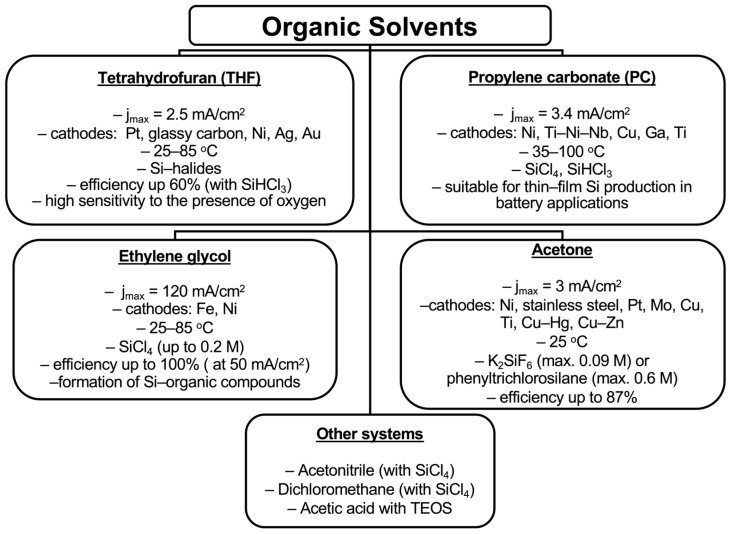
Overview of organic solvent systems for silicon electrodeposition and their typical process parameters.

**Table 1 materials-18-04009-t001:** Comparison of energy consumption and direct CO_2_ emissions for different silicon production processes.

Process	Energy Consumption (kWh/kg_Si_)	CO_2_ Emissions (kg_CO2_/kg_Si_)
Metallurgical route	10.5–13	11–13
Siemens process	44–70	20–30
Electrochemical deposition	7–9	1.6–3.1 ^1^

^1^ with carbon-free anodes, direct CO_2_ emissions can be eliminated.

**Table 2 materials-18-04009-t002:** Comparative summary of electrolyte systems for silicon electrodeposition.

Electrolyte Class	Max Current Density (A/cm^2^)	Operating Temp (°C)	Si Purity Potential	Advantages	Main Drawbacks
Fluoride-based melts	1.6–2	~550–1300	High (>99.9%)	High current efficiency possible; SiO_2_ dissolves well; high growth rates.	High T, hygroscopicity, anodic effect risk, difficult product washing.
Chloride-based melts	0.2–0.5	~300–1000	High (>99.9%)	Better water solubility; lower aggressiveness than fluorides.	Do not dissolve SiO_2_; Cl_2_ can evolve at anode.
Fluoride–chloride mixed melts	0.5–1.5	~600–850	High (>99.9%)	Improved practicality vs. pure fluorides.	Still high T; composition-dependent corrosion.
Oxide-additive molten salts	0.2–2 (composition-dependent)	Follows base melt; up to ~1370 °C	High (>99.9%)	Additives enhance SiO_2_ dissolution/transport; thicker films.	Process control; in BaCO_3_ system, high T (~1370 °C) is used.
Organic solvents	<0.01–0.01	~25–100	Moderate (oxide contamination common)	Low-T operation; easy handling.	Low j; impurity contamination.
Ionic liquids	<0.01–0.01	~25–120	Moderate (oxide contamination common)	Low-T; tunable chemistries.	Low j; oxide formation; moisture sensitivity.

## Data Availability

No new data were created or analyzed in this study. Data sharing is not applicable to this article.

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
