# Peer review of "Electrochemical Deposition of Silicon: A Critical Review of Electrolyte Systems for Industrial Implementation"

_materials, 2025, doi:10.3390/ma18174009_

Round 1

Reviewer 1 Report

Comments and Suggestions for Authors

Overall, the manuscript is well organized and informative, but I have several comments and questions that I hope will help strengthen the paper.

Major comments, questions and suggestions:

  1. Line 11–12: You write “This review analyzes the current state of research on electrolyte systems used for silicon electrodeposition…” Could you briefly state in the abstract how you evaluated those systems—e.g., “based on conductivity, stability, precursor solubility, etc.”?
  2. Line 29: You mention both the metallurgical and Siemens processes’ energy and CO₂ costs; a tabulated comparison in the introduction or as a supplementary table might help readers quickly grasp the magnitude of savings.
  1. Line 87–89: You note an “anodic effect” attributed to moisture in the melt . Have more recent studies quantified allowable water content or proposed strategies (e.g. in-line drying) to suppress this effect?
  2. Lines 116–118: Discussion of silver cathodes briefly cites the Ag–Si eutectic temperature. It might be useful to add a sentence on alternative noble‑metal substrates (e.g. platinum or gold) and their trade‑offs.
  1. Line 206–210: The 1959 patent by Stern & McKenna is interesting; however, details are sparse. If possible, could you include any follow‑up studies that revisit KCl–NaCl systems with modern analytic tools?
  2. Lines 241–247: For CaCl₂‑based melts, you mention unavoidable Si–Ca alloy formation . Could the authors discuss electrochemical or post‑treatment strategies to remove or convert that alloy back to pure Si?
  1. Line 305–307: You note that mixed melts “do not result in chlorine evolution at the anode.” Is that universally true, or does it depend on the exact Cl⁻/F⁻ ratio? A brief statement on the electrochemical window would be helpful.
  2. Lines 313–318: The role of SiO₂ additives is well covered, but it would strengthen the review to tabulate how solubility and grain size vary quantitatively with temperature and KCl/KF ratio.
  1. Line 449–450: You report 99.97 % Si purity at 1370 °C . Given that industrial plants rarely exceed 1200 °C, could you comment on whether similar purity is achievable at lower temperatures (e.g. via alternative oxide additives)?
  1. Line 539: The formation of SiO₂ films in ionic liquids is intriguing but problematic for pure Si production . Do any groups report truly elemental Si (i.e. < 1 % O) from ionic liquids, or is oxide always inevitable?
  2. Line 650–655: For organic solvents (e.g. propylene carbonate), high hydrogen content and limited film thickness remain key drawbacks. A succinct comparison of maximum achievable current densities across classes (fluoride vs. chloride vs. organic) would help readers see where ionic liquids/solvents stand quantitatively.
  1. Line 801–807: You outline ideal electrolyte traits. It may strengthen the conclusion to rank the top three candidate systems against those criteria in a summary table.
  2. Lines 882–886: The “ongoing debate” on one‑ vs. two‑step reduction mechanisms is noted, but perhaps you could suggest specific in situ techniques (e.g. Raman spectroscopy, electrochemical impedance spectroscopy) that future work should apply to resolve this.

Author Response

Dear Reviewer,

thank you very much for your valuabled comments and invested time. We are sending our answers in bold!

Overall, the manuscript is well organized and informative, but I have several comments and questions that I hope will help strengthen the paper.

Major comments, questions and suggestions:

Line 11–12: You write “This review analyzes the current state of research on electrolyte systems used for silicon electrodeposition…” Could you briefly state in the abstract how you evaluated those systems—e.g., “based on conductivity, stability, precursor solubility, etc.”?

These systems are evaluated based on key characteristics relevant to such implementation, including silicon precursor solubility, electrical conductivity, applicable current density, and behavior under process conditions.

Line 29: You mention both the metallurgical and Siemens processes’ energy and CO₂ costs; a tabulated comparison in the introduction or as a supplementary table might help readers quickly grasp the magnitude of savings.

Compared to the conventional metallurgical and Siemens processes, the electrochemical route requires substantially less energy and generates considerably lower direct CO emissions (Table 1) [1].

Table 1. Comparison of energy consumption and direct CO₂ emissions for different silicon production processes.

Process

Energy consumption (kWh/kgSi)

CO₂ emissions (kgCO₂/kgSi)

Metallurgical route

10.5 – 13

11 – 13

Siemens process

44 – 70

20 – 30

Electrochemical deposition

7 – 9

1.6 – 3.11

1 with carbon-free anodes, direct CO₂ emissions can be eliminated.

Line 87–89: You note an “anodic effect” attributed to moisture in the melt . Have more recent studies quantified allowable water content or proposed strategies (e.g. in-line drying) to suppress this effect?

There are no recent studies investigating the anodic effect caused by moisture in melts, and no further articles have addressed this topic since the 1980s.

Lines 116–118: Discussion of silver cathodes briefly cites the Ag–Si eutectic temperature. It might be useful to add a sentence on alternative noble‑metal substrates (e.g. platinum or gold) and their trade‑offs.

Platinum and gold have also been investigated as cathode materials. Platinum exhibits high chemical stability in halide melts; however, silicon readily forms platinum silicides through solid-state reactions at approximately 200–600 °C. At the elevated temperatures typical of molten halide baths (600–850 °C), extensive silicide formation and silicon diffusion into platinum are therefore expected, which may lead to alloying, contamination of the deposit, and complications in cathode reuse. Gold is even less suitable for high-temperature applications: the Au–Si system possesses a low-melting eutectic at 363 °C (~19 at.% Si), meaning that at typical deposition temperatures a liquid Au–Si phase forms, which readily dissolves silicon and promotes alloying.

Line 206–210: The 1959 patent by Stern & McKenna is interesting; however, details are sparse. If possible, could you include any follow‑up studies that revisit KCl–NaCl systems with modern analytic tools?

Unfortunately, to the best of our knowledge, no rigorous modern studies have revisited the binary NaCl–KCl system for silicon electrodeposition using advanced electrochemical or structural characterization techniques.

Lines 241–247: For CaCl₂‑based melts, you mention unavoidable Si–Ca alloy formation . Could the authors discuss electrochemical or post‑treatment strategies to remove or convert that alloy back to pure Si?

The alloy was then treated in an aqueous NHCl solution, which dissolved calcium as CaCl and released hydrogen, leaving silicon intact [38]. Other reported methods include leaching with dilute acids, thermal oxidation followed by leaching, and electrochemical techniques such as pulse potentials or using substrates with low calcium affinity to reduce alloy formation.

Line 305–307: You note that mixed melts “do not result in chlorine evolution at the anode.” Is that universally true, or does it depend on the exact Cl⁻/F⁻ ratio? A brief statement on the electrochemical window would be helpful.

They exhibit high current density, have low melting temperatures characteristic of alkali metal chlorides, effectively dissolve silicon dioxide, and, in fluoride-rich compositions, suppress chlorine evolution at the anode by widening the anodic electrochemical window. In contrast, chloride-rich mixtures narrow this window, making Cl2 release possible if the anodic limit is exceeded. For example, the LiCl–KCl eutectic has an electrochemical window of approximately 3.67 V at 773 K.

Lines 313–318: The role of SiO₂ additives is well covered, but it would strengthen the review to tabulate how solubility and grain size vary quantitatively with temperature and KCl/KF ratio.

While our review discusses the influence of temperature and KCl/KF ratio on SiO solubility and resulting silicon grain size, the quantitative data available in the literature are limited and often reported under different experimental conditions, making direct comparison challenging.

Line 449–450: You report 99.97 % Si purity at 1370 °C . Given that industrial plants rarely exceed 1200 °C, could you comment on whether similar purity is achievable at lower temperatures (e.g. via alternative oxide additives)?

The 99.97% purity figure refers specifically to the results of the cited study using barium carbonate as an additive, where the process temperature (~1370 °C) was sufficient to exceed the decomposition point of BaCO (~1360 °C). As noted in the manuscript, other oxide additives such as lithium oxide or calcium oxide have also been reported for operation at lower process temperatures, although their impact on purity at sub-1200 °C remains system-dependent.

Line 539: The formation of SiO₂ films in ionic liquids is intriguing but problematic for pure Si production . Do any groups report truly elemental Si (i.e. < 1 % O) from ionic liquids, or is oxide always inevitable?

To the best of our knowledge, no studies have reported truly oxygen-free silicon (< 1 wt% O) obtained directly from ionic liquids. Most works note the formation of SiO-rich surface layers or mixed Si/SiO phases, attributed to residual moisture, oxygen-containing impurities, or post-deposition oxidation. While surface oxide can be removed by chemical etching, oxide formation during electrodeposition in ionic liquids appears unavoidable under currently reported conditions.

Line 650–655: For organic solvents (e.g. propylene carbonate), high hydrogen content and limited film thickness remain key drawbacks. A succinct comparison of maximum achievable current densities across classes (fluoride vs. chloride vs. organic) would help readers see where ionic liquids/solvents stand quantitatively.

Line 801–807: You outline ideal electrolyte traits. It may strengthen the conclusion to rank the top three candidate systems against those criteria in a summary table.

A comparison of all electrolyte systems discussed is presented in Table 2.

Table 2. Comparative summary of electrolyte systems for silicon electrodeposition.

Electrolyte Class

Max Current

Density (A/cm²)

Operating Temp (°C)

Si Purity Potential

Advantages

Main Drawbacks

Fluoride-based melts

1.6–2

~550–900

High (>99.9%)

High current

efficiency possible; SiO₂ dissolves well; high growth rates.

High T,         hygroscopicity,  anodic effect risk, difficult product washing.

Chloride-based melts

0.2–0.5

~500–850

High (>99.9%)

Better water    solubility; lower aggressiveness than fluorides.

Do not dissolve SiO₂; Cl₂ can evolve at anode.

Fluoride–Chloride mixed melts

0.5–1.5

~500–850

High (>99.9%)

Improved     practicality vs pure fluorides.

Still high T;    composition     dependent      corrosion.

Oxide-additive molten salts

0.2–2 (composition-dependent)

Follows base melt; up to ~1370 °C

High (>99.9%)

Additives enhance SiO₂ dissolution/transport; thicker films.

Process control; in BaCO₃ system high T (~1370 °C) used.

Organic solvents

<0.01–0.01

~25–100

Moderate (oxide contamination common)

Low-T operation; easy handling.

Low j; impurity       contamination.

Ionic liquids

<0.01–0.01

~25–120

Moderate (oxide contamination common)

Low-T; tunable chemistries.

Low j; oxide    formation;    moisture       sensitivity.

Lines 882–886: The “ongoing debate” on one‑ vs. two‑step reduction mechanisms is noted, but perhaps you could suggest specific in situ techniques (e.g. Raman spectroscopy, electrochemical impedance spectroscopy) that future work should apply to resolve this.

As noted in the manuscript, in situ Raman spectroscopy, electrochemical impedance spectroscopy, and high-temperature XRD have already been applied to investigate the Si reduction mechanism in molten salts. These studies provide valuable insights but are typically limited to specific electrolyte compositions and temperature ranges, and thus have not yet resolved the one- vs. two-step debate universally. Future work combining these techniques—potentially supplemented by Si K-edge XAS—across a broader range of systems would help establish the mechanism more definitively.

I hope our answers can fully improve our ppaer.

Reviewer 2 Report

Comments and Suggestions for Authors

The manuscript provides a broad and well-structured review of the literature on electrochemical silicon deposition from molten salts, with a focus on industrial applicability. It covers the historical development of the topic (since 1865), categorizes electrolyte systems (fluoride-based, chloride-based, mixed, organic, oxide-containing), and analyzes their advantages and limitations. The authors point out research gaps and outline directions for future studies, which is a key requirement for MDPI review articles. After minor corrections and additions, the article can be printed:

  • Review methodology section – missing explanation of how the literature was selected (databases used, time range, exclusion criteria). MDPI expects transparency here.
  • Graphical/Tabular elements – while figures summarize some data (e.g., overview of organic and chloride–fluoride systems), more comparative tables summarizing all electrolytes with parameters (current density, temperature, purification challenges) would be valuable.
  • Conclusions section – could be more focused on future research challenges and clearer recommendations for industrial implementation.
  • Language and structure – overall readable, but some paragraphs are excessively long and could be broken up for better flow.
  • Missing graphical abstract – strongly recommended for MDPI, especially for review papers.

Author Response

Dear Reviewer,

thank you very much for valuabled comments and invested time. We are sending our answers in bold.

The manuscript provides a broad and well-structured review of the literature on electrochemical silicon deposition from molten salts, with a focus on industrial applicability. It covers the historical development of the topic (since 1865), categorizes electrolyte systems (fluoride-based, chloride-based, mixed, organic, oxide-containing), and analyzes their advantages and limitations. The authors point out research gaps and outline directions for future studies, which is a key requirement for MDPI review articles. After minor corrections and additions, the article can be printed:

  • Review methodology section – missing explanation of how the literature was selected (databases used, time range, exclusion criteria). MDPI expects transparency here.

Yes! We included it in our improved version: Methodology and results.

  • Graphical/Tabular elements – while figures summarize some data (e.g., overview of organic and chloride–fluoride systems), more comparative tables summarizing all electrolytes with parameters (current density, temperature, purification challenges) would be valuable.

A comparison of all electrolyte systems discussed is presented in Table 2.

Table 2. Comparative summary of electrolyte systems for silicon electrodeposition.

Electrolyte Class

Max Current

Density (A/cm²)

Operating Temp (°C)

Si Purity Potential

Advantages

Main Drawbacks

Fluoride-based melts

1.6–2

~550–900

High (>99.9%)

High current

efficiency possible; SiO₂ dissolves well; high growth rates.

High T,         hygroscopicity,  anodic effect risk, difficult product washing.

Chloride-based melts

0.2–0.5

~500–850

High (>99.9%)

Better water    solubility; lower aggressiveness than fluorides.

Do not dissolve SiO₂; Cl₂ can evolve at anode.

Fluoride–Chloride mixed melts

0.5–1.5

~500–850

High (>99.9%)

Improved     practicality vs pure fluorides.

Still high T;    composition     dependent      corrosion.

Oxide-additive molten salts

0.2–2 (composition-dependent)

Follows base melt; up to ~1370 °C

High (>99.9%)

Additives enhance SiO₂ dissolution/transport; thicker films.

Process control; in BaCO₃ system high T (~1370 °C) used.

Organic solvents

<0.01–0.01

~25–100

Moderate (oxide contamination common)

Low-T operation; easy handling.

Low j; impurity       contamination.

Ionic liquids

<0.01–0.01

~25–120

Moderate (oxide contamination common)

Low-T; tunable chemistries.

Low j; oxide    formation;    moisture       sensitivity.

  • Conclusions section – could be more focused on future research challenges and clearer recommendations for industrial implementation.

Thus, from an industrial perspective, mixed fluoridechloride melts containing KSiF and SiOparticularly KFKCl systemscurrently offer the most balanced combination of high current density, effective SiO dissolution, and suppressed chlorine evolution. Tungsten and molybdenum substrates have shown the best stability under these conditions. However, existing studies are limited to specific molar ratios (1:1 and 2:1), and comprehensive data on achievable silicon purity remain scarce. Compared with organic solvents and ionic liquids, which are constrained by low current densities and limited scalability, mixed fluoride–chloride systems appear more viable for metallurgical- and polycrystalline-grade silicon production in the near term.

Several open questions remain in this field. These include the influence of tempe-rature, substrate material, precursor concentration, and the role of chloride and fluoride cations on the kinetics of silicon deposition. While some studies have investigated dif-ferent metal cations, most experiments used various unstable silicon salts, without sys-tematically analyzing how chloride and fluoride ions affect the kinetics of silicon electrodeposition from KSiF.

Despite nearly 175 years of research, no consensus has been reached regarding the fundamental nature of the electrochemical process. There is ongoing debate over whether the reduction is reversible, quasi-reversible, or irreversible, and whether it occurs through a single-step or two-step mechanism. Addressing these challenges through systematic experimental work, coupled with advanced in situ diagnostics, will be crucial to optimizing process parameters, minimizing unwanted alloy formation, and identifying electrolyte systems capable of scalable, low-emission silicon production.

  • Language and structure – overall readable, but some paragraphs are excessively long and could be broken up for better flow.

Generally. language will be checked and improved in final step by technical Editor. We will improve it in our proofreading in final step!

  • Missing graphical abstract – strongly recommended for MDPI, especially for review papers.

You have right, but a writing of graphical abstract is only one choice by each author, and not our obligation. Therefore, we will not prepare a graphical abstract. I hope that our readers have enough information to understand our review paper.

Reviewer 3 Report

Comments and Suggestions for Authors
  1. The abstract provides a structured overview of multiple electrolyte systems. Key parameters (e.g., conductivity, temperature range, precursor solubility) are also addressed. However, it does not include any numerical ranges for conductivity, temperature, or deposition current density.
  2. The Introduction section provides a thorough historical overview, referencing developments from 1865 to the 1990s. It mentions critical progress points, such as the introduction of molten salts, the development of Ag cathodes, and the economic synthesis of K2SiF6. However, the summary omits recent developments in ionic liquids, Deep eutectic solvents (DESs), Solid-state electrolytes, and Hybrid molten/organic systems.
  3. The section accurately identifies the gap: insufficient clarity regarding which electrolyte systems are scalable or viable for industrial application. The review articulates its objective and rationalizes the omission of an in-depth discussion on morphology and purity, which is suitable for a process-oriented review. The final paragraph should delineate the paper's organization by comparing various systems according to thermodynamic, kinetic, and process parameters.
  4. Section 2, particularly subsection 2.1, is currently organized chronologically rather than thematically, leading to redundancy and difficulty comparing results across systems. I recommend reorganizing it into subthemes, such as electrochemical behavior and mechanisms, influence of cathode materials, fluoride system variants, alternative silicon precursors, and process efficiency and impurity control.
  5. Additionally, while many individual studies are cited, the reader is left to compare all the data mentally, which undermines the “critical review” aspect. I recommend including a summary table listing electrolyte composition, temperature (°C), Si source, cathode material, current density (mA/cm²), efficiency, etc. Please ensure that all relevant subsections are also checked.
  6. The conclusion reiterates earlier content and does not propose clear future research directions. Therefore, I recommend reframing the conclusion to summarize key findings succinctly, identify specific knowledge gaps, and recommend targeted experiments or computational studies to close those gaps.

Author Response

Dear Reviewer,

thank you very much for your valuabled comments and invested time. We are sending our answers in bold letters.

  1. The abstract provides a structured overview of multiple electrolyte systems. Key parameters (e.g., conductivity, temperature range, precursor solubility) are also addressed. However, it does not include any numerical ranges for conductivity, temperature, or deposition current density.

We have improved the abstract by adding this information: Fluoride-based melts offer high current densities (up to 2 A/cm²) and effective SiO dis-solution, but operate at high temperatures (600–1300 °C) and suffer from hygroscopicity. Chloride systems exhibit lower operating temperatures (300–1000 °C) and better water solubility but lack compatibility with common silicon sources. Mixed fluoride-chloride electrolytes emerge as the most promising option, combining high performance with improved practicality; they operate at 600–800 °C and current densities up to ~1.5 A/cm².

The Introduction section provides a thorough historical overview, referencing developments from 1865 to the 1990s. It mentions critical progress points, such as the introduction of molten salts, the development of Ag cathodes, and the economic synthesis of K2SiF6. However, the summary omits recent developments in ionic liquids, Deep eutectic solvents (DESs), Solid-state electrolytes, and Hybrid molten/organic systems.

Thank you for the comment. Ionic liquids are already discussed in the manuscript (Section 2.5). To the best of our knowledge, there are currently no published studies reporting the use of deep eutectic solvents (DESs), hybrid molten/organic systems, or solid-state electrolytes for the electrochemical electrodeposition of silicon. Solid-state electrolytes have only been applied in solid-oxide-membrane electrolysis, which is an electro-deoxidation process rather than electrodeposition.

The section accurately identifies the gap: insufficient clarity regarding which electrolyte systems are scalable or viable for industrial application. The review articulates its objective and rationalizes the omission of an in-depth discussion on morphology and purity, which is suitable for a process-oriented review. The final paragraph should delineate the paper's organization by comparing various systems according to thermodynamic, kinetic, and process parameters.

A comparison of all electrolyte systems discussed is presented in Table 2.

Table 2. Comparative summary of electrolyte systems for silicon electrodeposition.

Electrolyte Class

Max Current

Density (A/cm²)

Operating Temp (°C)

Si Purity Potential

Advantages

Main Drawbacks

Fluoride-based melts

1.6–2

~550–900

High (>99.9%)

High current

efficiency possible; SiO₂ dissolves well; high growth rates.

High T,         hygroscopicity,  anodic effect risk, difficult product washing.

Chloride-based melts

0.2–0.5

~500–850

High (>99.9%)

Better water    solubility; lower aggressiveness than fluorides.

Do not dissolve SiO₂; Cl₂ can evolve at anode.

Fluoride–Chloride mixed melts

0.5–1.5

~500–850

High (>99.9%)

Improved     practicality vs pure fluorides.

Still high T;    composition     dependent      corrosion.

Oxide-additive molten salts

0.2–2 (composition-dependent)

Follows base melt; up to ~1370 °C

High (>99.9%)

Additives enhance SiO₂ dissolution/transport; thicker films.

Process control; in BaCO₃ system high T (~1370 °C) used.

Organic solvents

<0.01–0.01

~25–100

Moderate (oxide contamination common)

Low-T operation; easy handling.

Low j; impurity       contamination.

Ionic liquids

<0.01–0.01

~25–120

Moderate (oxide contamination common)

Low-T; tunable chemistries.

Low j; oxide    formation;    moisture       sensitivity.

Section 2, particularly subsection 2.1, is currently organized chronologically rather than thematically, leading to redundancy and difficulty comparing results across systems. I recommend reorganizing it into subthemes, such as electrochemical behavior and mechanisms, influence of cathode materials, fluoride system variants, alternative silicon precursors, and process efficiency and impurity control.

Thank you for your recommendation, but we can accept your reccomendation. I hope that our available structure is also suitable for our readers in order to understand it! 

Additionally, while many individual studies are cited, the reader is left to compare all the data mentally, which undermines the “critical review” aspect. I recommend including a summary table listing electrolyte composition, temperature (°C), Si source, cathode material, current density (mA/cm²), efficiency, etc. Please ensure that all relevant subsections are also checked.

We appreciate the suggestion. The requested parameters — including electrolyte composition, temperature, Si source, cathode material, current density, and efficiency — are already compiled in the manuscript in graphical form (Figures 1–6). These figures integrate data from all relevant subsections to allow direct comparison between studies. If preferred, we can additionally provide this information in tabular format for clarity.

The conclusion reiterates earlier content and does not propose clear future research directions. Therefore, I recommend reframing the conclusion to summarize key findings succinctly, identify specific knowledge gaps, and recommend targeted experiments or computational studies to close those gaps.

Addressing these challenges through systematic experimental work, coupled with advanced in situ diagnostics, will be crucial to optimizing process parameters, minimizing unwanted alloy formation, and identifying electrolyte systems capable of scalable, low-emission silicon production, while complementary computational modeling can provide predictive insight into thermodynamic stability, kinetic pathways, and process scalability. This combined approach will enable the identification of the most promising electrolyte compositions and operating conditions for industrial implementation.

Round 2

Reviewer 1 Report

Comments and Suggestions for Authors

After revision, this manuscript can be recommended for publication.